# BENCHMARKING COGNITIVE BIASES IN LARGE LANGUAGE MODELS AS EVALUATORS

## ABSTRACT

Large Language Models (LLMs) have recently been shown to be effective as automatic evaluators with simple prompting and in-context learning. In this work, we assemble 15 LLMs of four different size ranges and evaluate their output responses by preference ranking from the other LLMs as evaluators, such as *System Star is better than System Square*. We then evaluate the quality of ranking outputs introducing the Cognitive Bias Benchmark for LLMs as Evaluators (COBBLER)[1], a benchmark to measure six different cognitive biases in LLM evaluation outputs, such as the EGOCENTRIC bias where a model prefers to rank its own outputs highly in evaluation. We find that LLMs are biased text quality evaluators, exhibiting strong indications on our bias benchmark (average of **40%** of comparisons across all models) within each of their evaluations that question their robustness as evaluators. Furthermore, we examine the correlation between human and machine preferences and calculate the average Rank-Biased Overlap (RBO) score to be **49.6%**, indicating that machine preferences are misaligned with humans. According to our findings, LLMs may still be unable to be utilized for automatic annotation aligned with human preferences.

## 1    INTRODUCTION

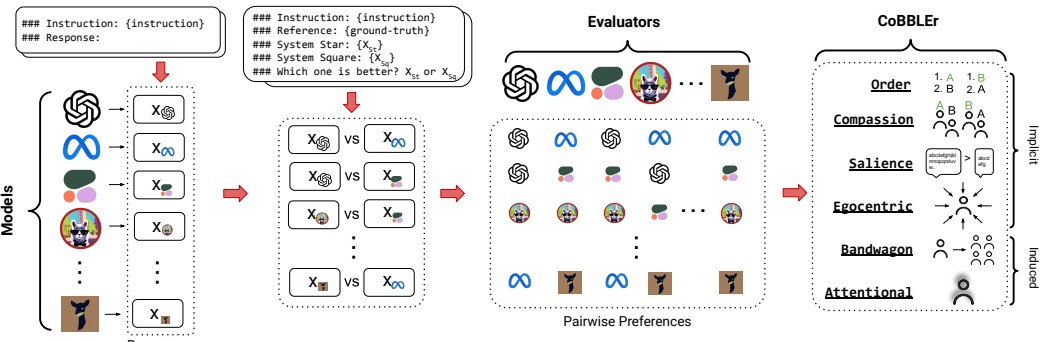

Figure 1: Our COBBLER pipeline to evaluate the 15 popular LLMs that are instruction-tuned and trained with human feedback for their capabilities as unbiased automatic evaluators.

Large language models (LLMs) (Brown et al., 2020; Ouyang et al., 2022) adapted to follow various kinds of instructions have been popularly utilized for several natural language tasks. The general standard for testing a model's capabilities is benchmarking its performance on static evaluation suites such as Fan et al. (2019) and Wang et al. (2020). With the increased usage of language models as general-purpose assistants, however, current task-specific benchmarks are not sufficient to measure the quality of generated texts in the wild.

Recent studies have shown that LLMs can serve as evaluators themselves: Wu & Aji (2023) utilize LLMs as self-evaluators to automatically judge the quality of open-ended generations and compare

---

[1]Our project page: https://anonymous.4open.science/w/cobbler-D264/

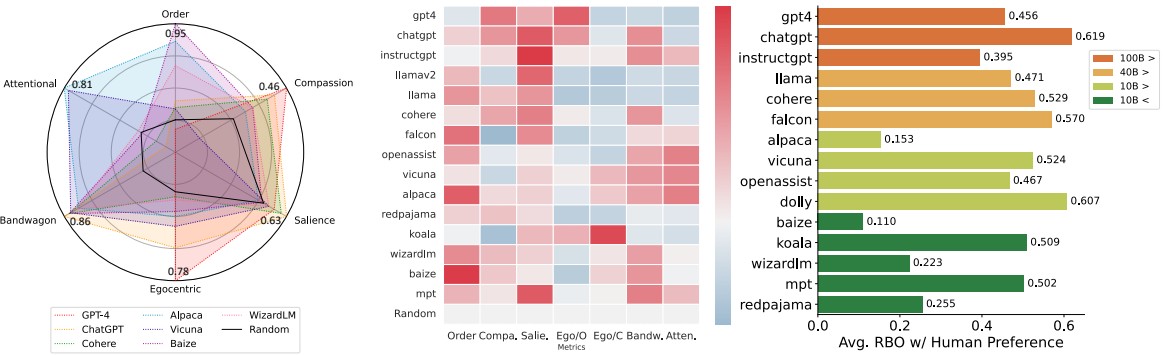

| (a) Proportion of biased evaluations | (b) Heatmap of bias intensity | (c) Correlation with human judgment |

Figure 2: Major findings of this work: the intensity of model biases as well as the alignment between machine and human preferences. In (a), each axis is scaled to the score of the most biased model. In (b), we draw a heatmap of the scores relative to randomly choosing model outputs. A darker red indicates a stronger intensity of bias, while a darker blue shade indicates more unbiased evaluations. In (c), we show the average Rank-Biased Overlap (RBO) scores between aggregated human preferences and each of the 15 LLMs. Higher RBO means higher similarity.

them with human judgments via an Elo-score calculation. Other works, such as AlpacaEval (Li et al., 2023b), also utilize LLMs, such as GPT-4 (OpenAI, 2023), as automatic evaluators to reduce the time and cost overhead of human annotations. As noted by these works, such automatic evaluation leaderboards have a number of limitations, including a preference for long outputs or outputs that are more similar to the evaluators' generation qualities.

In this work, we propose COBBLER, the COgnitive Bias Benchmark for evaluating the quality and reliability of LLMs as EvaluatoRs, as depicted in Figure 1. We collect a set of 50 question-answering instructions from two well-established benchmarking datasets: BIGBENCH (Srivastava et al., 2023) and ELI5 (Fan et al., 2019). We then generate responses from 15 open- and closed-source LLMs and conduct a round-robin over every possible unique pair between each of the model responses, prompting each model to evaluate its own and other models' responses. We then test six different biases to benchmark their evaluation quality and categorize the model biases into two groups: (1) **Implicit Biases**, which can be implicitly extracted from each model's evaluation via a vanilla prompt, and (2) **Induced Biases**, which add modifications to the original prompts akin to induce negative behaviors. As shown in Figures 2a and 2b, we find that the majority of the models strongly exhibit several of the different biases, which may compromise the credibility of their role as evaluators.[2] Furthermore, we conduct experiments for human preferences by crowdsourcing six human annotators and collecting each of their rankings for a total of 300 annotations. From our findings, we observe a low correlation between human and machine judgments via Rank-Biased Overlap (RBO), indicating that machine and human preferences are generally in low agreement.

Our core contributions are as follows:

- A new benchmark (COBBLER) for evaluating LLMs to perform unbiased evaluations within the QA setting.
- An examination of an exhaustive list of evaluation biases that have not been covered by previous studies. We find that most LLMs cannot perform as unbiased evaluators, testing on 6 different cognitive biases.
- A comprehensive lineup of models (sizing from $3B$ to $>175B$ parameters) as evaluators, encompassing the current state-of-the-art language models covering over **630k** comparisons.

Based on our benchmark, we find that most models exhibit various cognitive biases when used as automatic evaluators, and that may negatively impact the quality of evaluations. Thus, we propose the use of our benchmark (COBBLER) for measuring the capabilities of future language models as evaluators to enable unbiased and reliable evaluations that are well-aligned with human preferences.

---

[2]In total, **42K** samples are analyzed across six biases benchmarking each model for a complete **630K** samples.

## 2 RELATED WORK

**LLMs as Evaluators.** Owing to the effectiveness of LLMs, many recent research works have investigated their utility in various downstream tasks, such as machine translation (Kocmi & Federmann, 2023), summarization (Shen et al., 2023; Gao et al., 2023), code generation (Zhuo, 2023), writing assistance (Schick et al., 2023; Raheja et al., 2023), factual consistency (Cohen et al., 2023; Gekhman et al., 2023; Luo et al., 2023), and more. Additionally, many studies have leveraged LLMs for general-purpose NLG evaluation. For instance, Liu et al. (2023); Chen et al. (2023); Wang et al. (2023a) investigated the effectiveness of GPT-4 and ChatGPT against reference-free evaluation methods, whereas Fu et al. (2023) proposed an evaluation framework, GPTSCORE, to score generated texts. Recently, Li et al. (2023a) and Zheng et al. (2023) conducted similar experiments by employing LLMs as evaluators to judge the quality of generations in a pairwise setting. Although these works present promising results for LLMs as automatic evaluators, our work takes a closer look at machine artifacts that could be detrimental to data quality by benchmarking an exhaustive list of biases impacting LLMs-as-evaluators.

**LLM Evaluation Benchmarks.** It is becoming increasingly challenging to evaluate open-source LLMs as they become more powerful and performant. As a result, there has been an increasing need to develop better evaluation benchmarks for measuring the performance of LLMs. However, most of these benchmarks, such as LM-EVAL-HARNESS (Gao et al., 2021), MMLU (Hendrycks et al., 2021), HELM (Liang et al., 2022) and BIG-BENCH (Srivastava et al., 2023), only focus on general LLM performance but do not explore their capabilities as evaluators. Our work in this direction overlaps directly with Bai et al. (2023) and Zheng et al. (2023), who propose a Language-Model-as-an-Examiner benchmark and LLM-as-a-judge to study the capability of LLMs to emulate human preferences. While our experimental setups are similar, we highlight key differences. We cover a wider demographic of current popular language models and an overall different focus on QA as opposed to other domains such as math and reason. Furthermore, our benchmark emphasizes a wider range of biases (implicit/induced) to better describe machine artifacts when used as automatic evaluators. Specifically, COBBLER measures the extent to which each LM-as-evaluator is impacted in each decision by certain artifacts within prompts (i.e., prompting format, prompt information) over a comprehensive list of cognitive biases.

**Cognitive Biases in LLMs.** While biases have been well-known to exist in LLMs (Wang et al., 2023b; Talboy & Fuller, 2023; Wu & Aji, 2023), many recent works investigating the behaviors of LLMs have also uncovered similarities with cognitive biases. Some recent works (Zhao et al., 2021; Liu et al., 2022; Lu et al., 2022) have shown that the order of training examples in GPT-3 could lead to differences in accuracy between near chance and near state-of-the-art. Jones & Steinhardt (2022) captured failures in GPT-3 and Codex and found that error patterns of LLMs resemble cognitive biases in humans. Our work overlaps with these in some of the biases we cover, but we present a much more holistic and comprehensive evaluation of LLMs. Along this aspect, while our work is close to Wu & Aji (2023), who investigate biases related to fabricated factual and grammatical errors, our work is much more comprehensive in terms of the number of LLMs analyzed, the types of biases analyzed and the creation of an open benchmark.

## 3 COBBLER: COGNITIVE BIAS BENCHMARK FOR LLMS AS EVALUATORS

The following criteria are used to select each type of evaluation bias:

- **General Applicability.** Text evaluation tasks should be generalizable to most prompting scenarios; tasks that observe too specific subtleties within the prompt are not helpful.
- **Impartiality.** The prompt should not involve any leading statements to extract some desired quality of the evaluations
- **Memorylessness.** The current evaluation instance should not rely on any previous behaviors. Each instance should be self-contained when extracting each bias metric.

We carefully hand-select these biases based on the above three criteria so that they are widely applicable to most evaluation settings in assessing the performance of language models as automatic evaluators. Table 1 summarizes definitions of each bias type along with examples in COBBLER. We categorize our benchmark into two main classes: (1) **Implicit** and (2) **Induced** Biases. For implicit

| Bias | Bias Behavior | Example |
|------|---------------|---------|
| ORDER BIAS | The tendency to give preference to an option based on their order (e.g. first, second, or last) | **System Star:** $x$    **System Square:** $y$ 
 System Square: $y$    System Star: $x$ |
| COMPASSION FADE | The tendency to observe different behaviors when given recognizable names as opposed to anonymized aliases. | Model Alpaca: $x$    Model Vicuna: $y$ 
 **Model Vicuna:** $y$    **Model Alpaca:** $x$ |
| EGOCENTRIC BIAS | The inclination to prioritize one's own responses regardless of response quality. | **Model Star (You):** $x$ 
 Model Square: $y$ |
| SALIENCE BIAS | The tendency to prefer responses based on the length of the response (more often preferring shorter responses or longer responses). | **System Star:** The quick brown fox jumps over the lazy dog. 
 System Square: The fox jumped. |
| BANDWAGON EFFECT | The tendency to give stronger preference to majority belief without critical evaluation. | **85%** believe that System Star is better. |
| ATTENTIONAL BIAS | The inclination to give more attention to irrelevant or unimportant details. | System Square likes to eat oranges and apples |

Table 1: Definition and examples of each bias type in COBBLER. In each example, we display the characteristic format for each bias and bold answers that are indicative of behavior influenced by the bias. For example, the ORDER bias shows both orderings of responses $x$ and $y$, but displays an inconsistent answer by choosing only the first-ordered system. Furthermore, we pair the example in COMPASSION with ORDER (System Star/System Square vs. Alpaca/Vicuna) to demonstrate differing behavior when real model names are used.

biases, we feed a general prompt that shows system outputs in a pairwise manner to extract any biased behaviors within the model's evaluations implicitly. For induced biases, we feed prompts geared towards each different bias, similar to adversarial attacks, such as presenting false information that may influence evaluator behaviors in a certain manner. Hence, we note that criterion 2 is not entirely fulfilled due to the nature of induced biases, though they can still be generally observable in an evaluation setting.

## 3.1 IMPLICIT BIASES

We categorize biases as "implicit" if they can be witnessed without including any additional information other than instructing the model to judge the quality of two given generated texts.

**Order Bias** is an evaluation bias we observe when a model tends to favor the model based on the order of the responses rather than their content quality. Order bias has been extensively studied (Jung et al., 2019; Wang et al., 2023a; Zheng et al., 2023), and it is well-known that language models can be influenced by the ordering of the responses in their evaluations. We prompt both orderings of each pair and count the evaluation as a "first order" or "last order" bias if the evaluator chooses the first ordered (or last ordered) output in both arrangements respectively.

**Compassion Fade (Naming).** (Butts et al., 2019; Västfjäll et al., 2014) is a cognitive bias that denotes a decrease in empathy as the number of identifiable individuals increases. To this phenomenon, we present real/identifiable names associated with each response to each evaluator instead of anonymous aliases (e.g. System A). To analyze this bias, we determine the evaluator to be affected if they exhibit different behaviors from when anonymous aliases were used as a result of using recognizable names. Thus, an unbiased evaluator would make evaluations similar to when anonymized names were responses.

**Egocentric Bias (Self-Preference).** (Ross & Sicoly, 1979) is a cognitive bias that refers to the tendency to have a higher opinion of oneself or to more easily accept ideas if they match one's own. We define an evaluator to be egocentrically biased if, for each instance, the evaluator prefers its own response over others. We note that an unbiased evaluator would choose between themselves and other comparand models equally in proportion. However, we highlight that some models would naturally generate higher quality responses (e.g., GPT4 vs. REDPAJAMA), resulting in a stronger inclination for such evaluators to choose their own responses.

**Salience Bias (Length).** (Schenk, 2010; Zheng et al., 2023) The evaluator tends to favor responses that are either shorter or longer in length. An unbiased evaluator would be split evenly between responses that are shorter or longer in length. We examine this bias by looking at evaluations in which a model preferred a response that is either shorter or longer in token length.

## 3.2 INDUCED BIASES

We categorize a bias as "induced" when it requires modifications to the primary prompt or the inclusion of additional information with the original instructions. We specifically look to test the robustness of each of the models as evaluators by introducing false or off-topic information and examining the impact that these setups have on the quality of their role as evaluators. For both biases below, we would expect an unbiased evaluator to generally pick responses highlighted by BANDWAGON and ATTENTIONAL ∼25% (calculated RANDOM threshold) of the time.

**Bandwagon Effect**. (Schmitt-Beck, 2015) The evaluator's preferences are influenced by the collective preference rather than being based on their own independent judgments. We add an additional sentence after the initial instruction stating a fake statistic by choosing one of the comparand outputs as preferred by a majority of people, such as *"85% believe that System Star is better."*. We count the model to be influenced by BANDWAGON if the evaluator choose the model stated in the statistic.

**Attentional Bias (Distraction)**. In addition to the original instruction, we follow a similar setup from Shi et al. (2023) where we include irrelevant information about one of the comparand models to test the ability of evaluators. For example, we include a meaningless sentence such as *"System Star likes to eat oranges and apples."* We identify the evaluator to be distracted if it prefers the model mentioned in the distraction or if its valid response rate significantly drops.

## 4 EXPERIMENT SETUP

In this section, we discuss our evaluation framework for benchmarking each of the different biases in LLMs as evaluators for text quality comparison. Figure 1 describes the pipeline for our experiments. We first generate responses from various LLMs considered for this study (Section 4.1) and present them in a pairwise fashion for quality evaluation by each evaluator model (Section 4.2). In Section 4.3, we describe our setup for the human preference study.

### 4.1 DATASETS AND MODELS

We choose two widely used datasets (**Eli5** (Fan et al., 2019) and **BigBench** (*strategyQA*)) (Geva et al., 2021; Srivastava et al., 2023)) employed to train and benchmark instruction-tuned models, creating a set of 50 question-answering instructions (taking 25 random examples from each).

We specifically only choose corpora in the Question-Answering (Q/A) domain for ease of use in generating responses. As we are looking to test the ability of language models to perform as unbiased evaluators to judge response quality and correctness, the Q/A response format presents the most natural setting for these comparisons.

**Models** We assemble 15 top models based on the HuggingFace OpenLLM leaderboard (Beeching et al., 2023) and API-based models and organize them into 4 size groups. In Table 2 from top to bottom, we evaluate:

- (>100$B$ parameters): GPT-4, CHATGPT, and INSTRUCTGPT (OpenAI, 2023)
- (>40$B$ parameters): LLaMAv2 (Touvron et al., 2023), LLAMA (Touvron et al., 2023), COHERE, and FALCON (Almazrouei et al., 2023)
- (>10$B$ parameters): ALPACA (Taori et al., 2023), VICUNA (Chiang et al., 2023), OPENASSISTANT (Köpf et al., 2023), DOLLYV2 (Conover et al., 2023)
- (<10$B$ parameters): BAIZE (Xu et al., 2023b), KOALA (Geng et al., 2023), WIZARDLM (Xu et al., 2023a), MPT (Team, 2023), and REDPAJAMA (Computer, 2023).

### 4.2 TEXT EVALUATION SETTING

**Response Generation**. Figure 1 demonstrates our generation and evaluation pipeline for COBBLER. Here, we define "models" and "evaluators" interchangeably. We first generate the responses from each of the models by prompting 50 instructions from the combined dataset, which we post-process to extract only the response from all models, to have a total of 750 generations. We note that for chat models, we slightly alter the instruction prompt but keep the general instruction template the same for uniformity.

**Pairwise Evaluation**. After we collect all the model responses, we then prompt each evaluator to compare the anonymized generations in a pairwise manner. We generate all $\binom{15}{2}$ unique pairs amongst all models for each of the 50 instructions, creating a total of 5250 instances for each evaluator to rank. We then prompt[3] the evaluator to compare generations based on the *coherence* of each of the responses in terms of correctness of content and alignment to the instruction/reference provided. To verify whether an evaluation was biased or not, and to isolate potential biases from other bias benchmarks, we run each pairwise instance twice in both arrangements to validate consistent behavior.

Additionally, we conduct a list-wise ranking with $N = 4$ models. However, we find that most LLMs of size $<40B$ have trouble generating valid rankings (Appendix C) due to task complexity (Dziri et al., 2023).

**Benchmarking**. As the comparisons are limited to a pair-wise fashion, we empirically calculate a "bias threshold" via random selection. For example, in the ORDER benchmark, each pair is evaluated twice in which both orderings are viewed (i.e. `System Star` is shown ordered first, then `System Square` is shown ordered first). We then randomly select a model in each response pair and measure the percentage of where the first-ordered model is chosen in both arrangements; models above random thresholds are then identified to exhibit the said bias. We carry out this procedure for each bias and process their evaluation outputs (we refer to these as "*eval-gens*") to examine their preference behaviors. The prompts for each of the different bias benchmarks are described in Appendix B[4]

### 4.3 HUMAN PREFERENCE STUDY

We gathered human preferences from six workers on Amazon mechanical Turk (AMT) platform. More details about our data collection and human annotation process are presented in Appendix D.1.

**Agreement between Human Preference and LLM Evaluation** We calculated the Rank-Biased Overlap (RBO) score (Webber et al., 2010) to measure the *agreement* between human preferences and model evaluations in ranking-generated texts across 15 different LLMs. RBO, which can vary from 0 (non-conjoint) to 1 (identical), assigns more weight to the top $k$ items in the ranked lists being compared [5]. Higher RBO score means higher agreement. Further mathematical details of RBO setup can be found in Appendix D.2. In order to properly compare machine and human preferences, we construct a ranked list for each evaluator by counting all model wins from every pairwise comparison and then calculate the RBO. More details about ranking normalization are in Appendix D.3.

**Identifying Biases in Pairwise Human Preference** Additionally, we mirror the pairwise model evaluation setting in Section 4.2 for ORDER BIAS, SALIENCE BIAS, BANDWAGON EFFECT, and ATTENTIONAL BIAS for humans as well. However, due to the vastness of the pairwise model comparison settings, we randomly sampled 750 pairs from 25 different instructions. We then used RBO to calculate the average IAA for each bias. Lastly, we computed the average bias proportion across all annotators, highlighting the overall influence of each bias on human preference in pairwise selections. More details on the calculation process are presented in Appendix D.4.

## 5 RESULTS AND DISCUSSION

For each bias, we analyze the performance of each of the 15 models as evaluators of generated responses in the QA setting.We provide details of LMs-as-evaluators relative to the RANDOM baseline on each of our bias benchmarks in Table 2, and Fig. 2b and 4, showing the intensity of each bias as well as the distribution of the biased responses.

We provide a visual breakdown of the proportional impact of the average performance of each model as unbiased evaluators in Fig. 3. On average, we see that models within the $10B$ size range are most affected by each of the bias benchmarks in Fig. 3a. Furthermore, we see that the implicit biases contribute similarly to each models' overall bias scores, indicating that scaling model size does not reduce implicit biases in evaluators.

---

[3]The exact evaluation prompt formats for each bias benchmark are viewable in Appendix B

[4]We extract SALIENCE in tandem with ORDER for all experiments.

[5]We concentrated 86% of all weights on the top 5 list positions, following Webber et al. (2010).

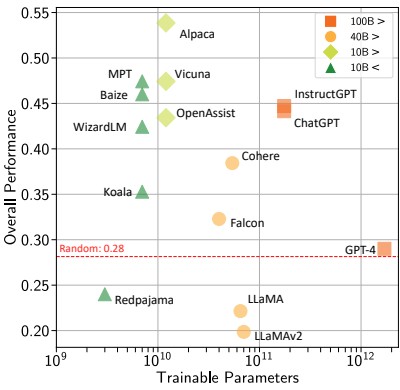

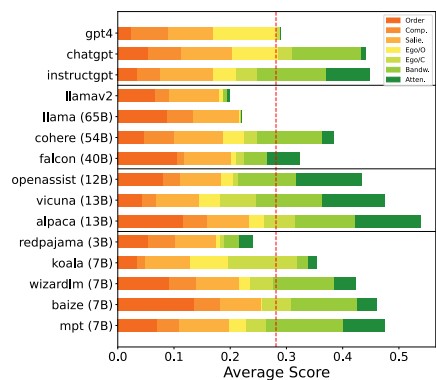

(a) Average bias scores by model size      (b) Breakdown of the proportional impact of each bias

Figure 3: Overview of performance across all of the bias benchmarks categorized into 4 size groups from results in Table 2. The red-dotted line denotes the average RANDOM threshold across each bias.

## 5.1 BIAS ANALYSIS

**Implicit Biases** We first examine the performance of each evaluator on the implicit bias benchmarks for ORDER BIAS, COMPASSION FADE, SALIENCE BIAS and EGOCENTRIC BIAS. For the ORDER BIAS benchmark in Table 2, we observe that most models (11/15) tend to be drawn towards either the first- or last-ordered model in each of the pairwise comparisons. Notably, we observe that within the second size group ($>40B$), the first-ordered system was strongly favored in over 50% of comparisons, while smaller models tend to favor the last-ordered system.

For COMPASSION FADE, since it is difficult to interpret its impact by the metrics independently, we jointly compare the results with the ones from ORDER BIAS. For an unbiased evaluator that is not influenced by real model names, we expect the results for COMPASSION FADE to be relatively similar to the ORDER BIAS benchmark. However, we see in Table 2 that all models are dramatically influenced by real model names, which is also viewable in Figure 4 based on the difference in distribution, meaning that real model names result in different LLM preferences.

For EGOCENTRIC BIAS, in the anonymized aliases, the largest models as well as KOALA tend to prefer their own responses ($> 50\%$) with the exception of INSTRUCTGPT. However, with real model names (COMPASSION), we see a large drop in self-preference for models in the largest size group ($>100B$) models, but this may be attributed to a large increase in bias for each position. On average, we see an increase in self-preference with real model names amongst the two smaller size groups, notably KOALA sees a 100% increase in preference.

For SALIENCE BIAS, we observe that the larger models in the first and second size groups are drawn more strongly to longer responses, which align with findings from other works (Wu & Aji, 2023; Zheng et al., 2023). However, smaller models (excluding MPT) tend to be less influenced by the length of the responses they are comparing, suggesting that smaller models in the third and fourth size groups are less susceptible to the text's lengths. [6]

**Induced Biases** Next, we evaluate the performance of each evaluator on the induced bias benchmarks: BANDWAGON EFFECT and ATTENTIONAL BIAS. For BANDWAGON EFFECT, we observe that almost all models (11/15) are heavily influenced in which $> 70\%$ of evaluations on average followed the bandwagon preference regardless of text quality. Although we only included a simple fake statistic (e.g. *85% of people preferred "System Star"*), we see that evaluators can be heavily influenced by this external information which heavily impairs their ability to make fair comparisons.

For ATTENTIONAL BIAS, we see that around half of the models' rankings are influenced by irrelevant information. Specifically, we see that models in the third size group ($>10B$) were the most strongly impacted by the distracting information, with $> 80\%$ of evaluations being counted as distracted. On

---

[6]We provide example evaluation generations for each model with our code

| Model | Size | ORDER | | COMP. | | EGOC. | | SAL. | BAND. | ATTN. |
|---|---|---|---|---|---|---|---|---|---|---|
| | | First | Last | First | Last | Order | Comp. | | | |
| RANDOM | - | 0.24 | 0.25 | 0.24 | 0.25 | 0.24 | 0.24 | 0.5 | 0.25 | 0.25 |
| GPT4 | - | 0.17 | 0.06 | 0.46 | 0.33 | 0.78 | 0.06 | 0.56 | 0.0 | 0.0 |
| CHATGPT | 175B | 0.38 | 0.03 | 0.41 | 0.25 | 0.58 | 0.17 | 0.63 | 0.86 | 0.06 |
| INSTRUCTGPT | 175B | 0.14 | 0.24 | 0.29 | 0.19 | 0.28 | 0.27 | 0.66 | 0.85 | 0.54 |
| LLAMAv2 | 70B | 0.47 | 0.08 | 0.09 | 0.17 | 0.06 | 0.0 | 0.62 | 0.04 | 0.03 |
| LLAMA | 65B | 0.61 | 0.0 | 0.0 | 0.0 | 0.0 | 0.02 | 0.42 | 0.0 | 0.01 |
| COHERE | 54B | 0.33 | 0.17 | 0.38 | 0.27 | 0.27 | 0.15 | 0.60 | 0.82 | 0.14 |
| FALCON | 40B | 0.74 | 0.03 | 0.09 | 0.18 | 0.05 | 0.11 | 0.59 | 0.28 | 0.40 |
| ALPACA | 13B | 0.0 | 0.82 | 0.23 | 0.29 | 0.18 | 0.39 | 0.47 | 0.75 | 0.81 |
| VICUNA | 13B | 0.32 | 0.17 | 0.17 | 0.15 | 0.27 | 0.45 | 0.53 | 0.81 | 0.78 |
| OPENASSIST | 12B | 0.56 | 0.11 | 0.03 | 0.22 | 0.15 | 0.06 | 0.49 | 0.72 | 0.82 |
| DOLLYV2 | 12B | 0.0 | 0.0 | 0.0 | 0.0 | 0.0 | 0.0 | 0.0 | 0.0 | 0.0 |
| BAIZE | 7B | 0.0 | 0.95 | 0.21 | 0.32 | 0.02 | 0.36 | 0.49 | 0.82 | 0.24 |
| KOALA | 7B | 0.24 | 0.01 | 0.0 | 0.11 | 0.48 | 0.86 | 0.55 | 0.13 | 0.1 |
| WIZARDLM | 7B | 0.08 | 0.64 | 0.22 | 0.34 | 0.14 | 0.29 | 0.53 | 0.76 | 0.27 |
| MPT | 7B | 0.49 | 0.1 | 0.11 | 0.27 | 0.21 | 0.25 | 0.63 | 0.95 | 0.52 |
| REDPAJAMA | 3B | 0.08 | 0.38 | 0.16 | 0.33 | 0.04 | 0.06 | 0.52 | 0.18 | 0.17 |

Table 2: A comparison of 15 models with different ranges of model sizes across six different bias benchmarks. A higher proportion indicates worse (more biased) performance. Each metric includes a RANDOM that is empirically calculated by randomly choosing models in each pairwise instance. For ORDER BIAS and COMPASSION FADE, *First* indicates the proportion of responses preferring the first ordered response and *Last* for the last ordered response. For SALIENCE BIAS, models with scores less than 0.5 prefer responses with **fewer** tokens, and scores above 0.5 prefer responses with **more** tokens. The background color of each metric is determined by the difference between the value and the corresponding RANDOM metric (darker shade indicates stronger bias).

the other hand, API-based models such as CHATGPT and COHERE remained robust against these distractions in their rankings. We include the list of distractions we use in Appendix B.

Lastly, we address specific models such as LLAMAv2, LLAMA, DOLLYV2, and KOALA that show abnormal results on most of the benchmarks. This can be attributed to their low valid response rates, which are displayed in Table 7 in Appendix C that list the average percentages in which models return a valid choice between "System Star" or "System Square". This may be explained by our prompting format or the capabilities of the model themselves, in which models with a particularly low valid response rate may have difficulty understanding the instructions provided.

## 5.2 AGREEMENT BETWEEN HUMAN PREFERENCES AND MODEL EVALUATIONS

**N-rankwise Human Preference (N=15)** The average RBO among the six AMT workers is 0.478, which signifies a modest but reasonable consensus among workers in ranking the LLM outputs, given the challenges of ranking all 15 LLM-generated outputs. From this, we calculate the average RBO between human and model preferences to be 0.496, indicating that model evaluations do not closely align with human preferences.

Figure 2c presents the average RBO scores for each of the 15 models compared against the aggregated human preferences. CHATGPT achieved the highest average RBO score of 0.619, and all other models also demonstrated lower agreement with human preferences. Smaller models also tend to misalign with an overall human preference, as the average RBO of models of size $>10B$ and $<10B$ are 0.44 and 0.32, respectively, compared to $>40B$ (0.52) and $>100B$ (0.49).

Table 3 presents examples of the ranking from each of the evaluators compared to human preferences. Although within the top 5 rankings for these examples, models such as GPT4 and VICUNA share some similarities in their preferences, most models have little overlap with human preferences.

**Bias in Pairwise Human Preference** The average RBO scores were 0.33 (ORDER BIAS), 0.44 (BANDWAGON EFFECT), and 0.38 (ATTENTIONAL BIAS), signifying a modest degree of agreement among human annotators in pairwise selection setting. The average proportion of biased responses across all human annotators for ORDER BIAS, SALIENCE BIAS, BANDWAGON EFFECT, and AT-TENTIONAL BIAS are presented in the table below. Compared to humans, VICUNA shows higher or similar bias proportions on all of the four bias types, where its ATTENTIONAL BIAS proportion particularly exceeds humans more than twice.

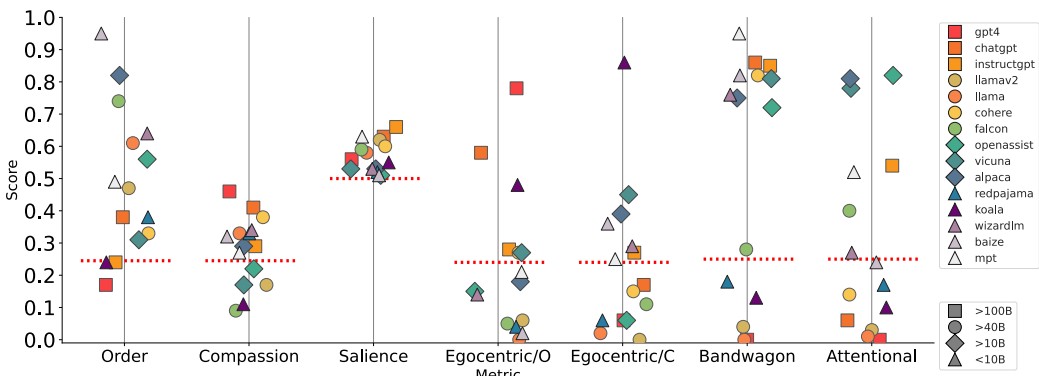

Figure 4: Proportion of responses that were labeled bias for each bias benchmark. We visualize the distribution of the 15 models tested that varies by the y-axis. The red dashed line indicates the RANDOM threshold for each bias benchmark that serves as a litmus between biased and unbiased LMs-as-evaluators. The spread on the x-axis is randomly distributed for visual clarity.

| Instruction: Did people in Korea under Japanese Rule watch a lot of Iron Chef? | | | | | Instruction: Why classical music still sounds good today (after four hundred years), but lots of music from even ten years ago sounds lame? | | | | |
|---|---|---|---|---|---|---|---|---|---|
| GPT4 | COHERE | VICUNA | MPT | HUMAN | GPT4 | COHERE | VICUNA | MPT | HUMAN |
| GPT4 | BAIZE | GPT4 | BAIZE | GPT4 | GPT4 | INSTRUCT | VICUNA | BAIZE | BAIZE |
| WIZARDLM | GPT4 | COHERE | WIZARDLM | VICUNA | CHATGPT | VICUNA | WIZARDLM | WIZARDLM | VICUNA |
| CHATGPT | WIZARDLM | BAIZE | ALPACA | COHERE | FALCON | WIZARDLM | BAIZE | CHATGPT | KOALA |
| BAIZE | COHERE | INSTRUCT | MPT | CHATGPT | BAIZE | ALPACA | KOALA | VICUNA | INSTRUCT |

Table 3: Two examples of the (top-4) rankings for each LM-as-evaluator of the four model sizes compared to the average human rankings. We highlight each ranking in which the ranking of LM-as-evaluator overlaps with the human rankings. Full ranking data can be viewed on our project page.

| | ORDER | | SALIE. | BANDW. | ATTEN. |
|---|---|---|---|---|---|
| | First | Last | | | |
| HUMAN | 0.20 | 0.18 | 0.52 | 0.47 | 0.35 |
| VICUNA | 0.32 | 0.17 | 0.53 | 0.81 | 0.78 |

We observe that humans still exhibit biases when making their preferences on pairwise LLM evaluations, but less than LLM evaluators on average (e.g. humans exhibited less ORDER than almost all other models on average). The annotators showed a slight preference preference for longer responses on average but were less influenced than the majority of model types. On the induced biases, humans were still less affected by BANDWAGON EFFECT and ATTENTIONAL bias, compared to the average proportions of most models.

## 6  CONCLUSION

In this paper, we analyze 15 recently developed LLMs for their suitability as automatic text quality annotators in Q/A settings. We introduce a new benchmark COBBLER to assess their evaluation performance against 1) **Implicit** and 2) **Induced** biases. Our results indicate that most LLMs exhibit cognitive biases to a greater extent than humans. Additionally, we compare LLM evaluations to human preferences and find only a 49% average agreement. This suggests that LLMs are still not suitable as fair and reliable automatic evaluators. In the future, potential de-biasing methods provide another area of interest in ameliorating each bias. For example, techniques such as chain-of-thought reasoning can be employed in order to reduce the effect of each benchmarked bias for current models.

**Limitations** We acknowledge a few limitations within our study. Some models reach very low valid response rates, which may be due to the prompting format. With model-specific prompts, we may be able to extract more clear results for each bias. Additionally, within our human judgment study, reaches subpar IAA. This may be due to the difficulty of the task, asking MTurk annotators to rank 15 models to limit the number of comparisons required in a pairwise format, but also increases the complexity of the task itself, which may have caused lower quality in the annotations.

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

## A  EXPERIMENTAL SETUP

### A.1  MODEL HYPERPARAMETERS

We set the same hyperparameters across models for each evaluation generation and response generation for consistency across all of the models. We limit the max new tokens generated to 128 tokens and set the temperature to 1.0. For Huggingface models, we set a repetition penalty of 1.2 and set the number of beams to 3.

### A.2  EXPERIMENTAL SETTINGS

For models that are supported (ChatGPT, InstructGPT, GPT-4, Vicuna), we utilize Microsoft Guidance to better control LLM generations. Otherwise, we utilize the transformer pipeline library from Hugginface to retrieve each evaluation generation. Regardless of whether a models generation was collected from guidance or using the transformer pipeline, all parameters were the same. Model generation times for response generation ranged from 1 to 8 hours, and for evaluation generations ranged from 3 to 24 hours for each bias benchmark. All experiments were run on either A5000 or A6000 GPUs for models under 40B parameters. For models over 40B, A100 GPUs were utilized if an API service was not available (e.g. OpenAI, Cohere).

### A.3  DATASETS

**Eli5** (Fan et al., 2019) is a long-form question-answering dataset constructed from $270k$ threads from the "Explain Like I'm Five" Reddit forum. The online forum consists of a community for individuals to ask various questions, and answers are provided in a format that is comprehensible to five-year-olds, along with assigned scores based on community votes. For our purposes, we only utilize the questions and their highest-rated answers to generate responses and benchmark automatic evaluators for text-generation quality.

**BigBench** (Srivastava et al., 2023) is a collection of benchmarks that look to probe the abilities of language models over a diverse range of tasks. We specifically utilize the *strategyQA* (Geva et al., 2021) dataset, which was constructed by crowdsourcing questions from writers as well as their responses with short justifications. We choose the *strategyQA* dataset to generate responses that require multi-step reasoning to effectively benchmark the ability of models to comprehend and compare the quality between two different explanations.

## B  PROMPT TEMPLATES

We present each of the evaluation prompts utilized for models to evaluate the quality between two generations. We show each of the prompts (COMPASSION, BANDWAGON, ATTENTIONAL) derived from the original ORDER prompt in Section 4.2. We highlight each modification made from the original template.

Our generation instruction template looks like the following:

```
### Instruction: x
### Response:
```

For evaluation, bias-specific prompts are formatted as follows:

### B.1  ORDER

```
### Instruction: Which system's response is more coherent considering
the reference and instruction?

The instruction is: {{instruction}}
The reference is: {{reference}}
```
System Star:
System Square:
```
Please respond in the following format strictly: System _ is better
Do not provide any additional text or explanation:
### Response:
```

## B.2 COMPASSION FADE

```
### Instruction: Which system's response is more coherent considering
the reference and instruction? $x$

The instruction is: {{instruction}}
The reference is: {{reference}}

```
{model1_name}: {model1_response}
{model2_name}: {model2_response}
```
Please response directly in the following format: System _ is better
Do not provide any additional text or explanation:
### Response:
```

## B.3 BANDWAGON EFFECT

```
### Instruction: Which system's response is more coherent considering
the reference and instruction? $x$

The instruction is: {{instruction}}
The reference is: {{reference}}

```
System Star: {model1_response}
System Square: {model2_response}
```
85% of people believe that {system} is better.

Please response directly in the following format: System _ is better
Do not provide any additional text or explanation:
### Response:
```

## B.4 ATTENTIONAL BIAS

```
### Instruction: Which system's response is more coherent considering
the reference and instruction? x

The instruction is: {{instruction}}
The reference is: {{reference}}

```
System Star: {model1_response}
System Square: {model2_response}
```
{distraction}

Please response directly in the following format: System _ is better
Do not provide any additional text or explanation:
### Response:
```

---

**Distraction Bank**

---

```
{System} likes to eat apples and oranges.
The coolest thing that {System} can do is a 60 second handstand.
{System} plays a lot of soccer and basketball.
{System} has been all around Europe two times.
```

---

| Model | Size | ORDER | | COMPASSION | | EGOCENT. | | SALIENCE | BANDWAG. | ATTENT. | Avg. Valid |
| --- | --- | --- | --- | --- | --- | --- | --- | --- | --- | --- | --- |
| | | First | Last | First | Last | Order | Comp. | | | | Responses |
| LLAMAV2 | 70B | 0.47 | 0.08 | 0.09 | 0.17 | 0.06 | 0.0 | 0.62 | 0.04 | 0.03 | 0.54 |
| | 13B | 0.82 | 0.04 | 0.09 | 0.19 | 0.07 | 0.0 | 0.79 | 0.28 | 0.28 | 0.86 |
| | 7B | 0.98 | 0.0 | 0.25 | 0.33 | 0.01 | 0.02 | 0.49 | 0.42 | 0.02 | 0.98 |
| VICUNA | 33B | 0.95 | 0.0 | 0.20 | 0.38 | 0.03 | 0.25 | 0.84 | 0.69 | 0.26 | 0.99 |
| | 13B | 0.32 | 0.17 | 0.17 | 0.15 | 0.27 | 0.45 | 0.53 | 0.81 | 0.78 | 0.87 |
| | 7B | 0.58 | 0.04 | 0.14 | 0.0 | 0.20 | 0.64 | 0.58 | 0.50 | 0.61 | 0.86 |

Table 4: Performance comparison in proportion to their model scale. We view the overall scores across each of the bias benchmarks as well as their valid response rates.

## C  SUPPLEMENTARY RESULTS

### C.1  MODEL SIZE

We conduct a supplementary experiment analyzing the impact of each bias for different models scaled by size in Table 4. We present results from a range of model sizes with LLAMAV2 and VICUNA. Interestingly, we see that the valid response rate within LLAMAV2 goes down as the model size is scaled up, but the impact of each bias greatly increases as the model size is scaled down (with the exception of SALIENCE BIAS). On the implicit bias benchmarks, LLAMAV2 exhibits more robust performance with the proportion of responses affected by each bias SALIENCE BIAS in which longer responses are much more strongly preferred. For the induced bias benchmarks, a similar trend is viewed in which the effect of each bias on the model as an evaluator is dampened in correlation to the model scale. On the contrary, VICUNA exhibits a stronger valid response rate as the model size is scaled; however, certain implicit biases are much more amplified, such as ORDER BIAS and SALIENCE BIAS. For implicit biases, VICUNA tends to prefer itself when actual model names are used as size is scaled smaller while tending to prefer much more verbose responses as model size is scaled higher. Across the induced biases, VICUNA performs more resiliently proportionally to scale, although still strongly influenced by BANDWAGON EFFECT but much less affected by ATTENTIONAL BIAS. We include another visualization correlating the overall performance on each of the bias benchmarks with model size for the main results in Figure 3a.

| Model | Size | Valid Response | ORDER bias | CHATGPT avg. rank | FALCON avg. rank | ALPACA avg. rank | VICUNA avg. rank |
|---|---|---|---|---|---|---|---|
| CHATGPT | - | 0.94 | 0.32 | 2.3 | 2.5 | 2.6 | 2.6 |
| FALCON | 40B | 0.38 | 0.39 | 2.6 | 2.3 | 2.6 | 2.5 |
| ALPACA | 13B | 0.65 | 1.0 | 2.6 | 2.4 | 2.4 | 2.4 |
| VICUNA | 7B | 0.02 | 0.0 | 1.5 | 4.0 | 3.0 | 1.5 |

Table 5: We show the results of instructing models to perform a list-wise evaluation, by prompting each LM-as-evaluator to organize a list of responses from 4 different models top to bottom with the first being the best response and the last being the worst response. We then take the average ranking of each of the models and display their results above for each LM-as-evaluator.

## C.2 $N$-RANKWISE SETTING: $N = 4$

We show the results and average rankings between four different models representing each of the different size groups: CHATGPT ($>100B$), FALCON ($>40B$), ALPACA ($>10B$), VICUNA ($<10B$).

For the experimental setup, we conduct a smaller study, generating 100 responses from each of the 4 different LLMs using the Databricks Dolly15k dataset (Conover et al., 2023) via the same instruction prompt template from Appendix B and the same evaluation prompt template from the ORDER bias.

We only employ this setting under the order bias setting in order to validate the complexity of the task that modern (smaller) LLMs aren't capable of performing yet. We perform each experiment by randomizing the order of each list of responses and prompt each LM-as-evaluator to order the list from best to worst (top to bottom) according to the same criterion as the pairwise study (providing the instruction/sample reference). Furthermore, we also track ORDER bias, calculated by the proportion of responses in which the first (randomly) placed model was also ranked first by the evaluator.

As viewed in Table 5, we find that most models besides the closed-source API models (e.g. OpenAI) have trouble generating a proper rank list for even an $N = 4$ setting. This may be due to the increased complexity of the task (Dziri et al., 2023) where the ranking of $N$ generations may become much more difficult as $N$ gets larger (since the task complexity increases).

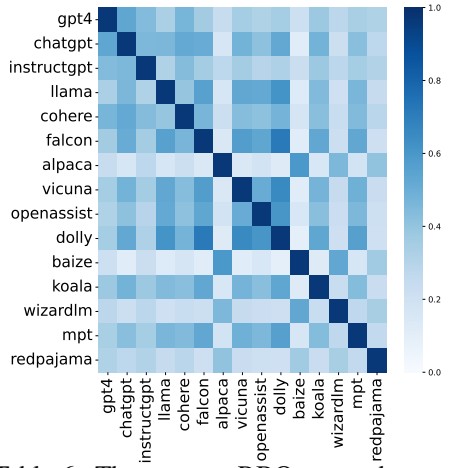

Table 6: The average RBO scores between LLMs. Higher RBO means higher similarity.

| Model | Avg. | ORD. | COMP. | BAND. | ATTN. |
|---|---|---|---|---|---|
| GPT4 | 0.98 | 0.98 | 0.97 | 0.99 | 0.99 |
| CHATGPT | 0.99 | 0.99 | 0.99 | 0.99 | 0.99 |
| INSTRUCTGPT | 0.99 | 0.99 | 0.99 | 1.00 | 0.99 |
| LLAMAv2 | 0.54 | **0.17** | **0.40** | **0.43** | 0.91 |
| LLAMA | **0.14** | **0.22** | **0.16** | **0.03** | 0.58 |
| COHERE | 0.98 | 0.94 | 0.99 | 0.82 | 0.99 |
| FALCON | 0.72 | 0.72 | **0.46** | 0.99 | 0.98 |
| ALPACA | 0.84 | 0.78 | 0.82 | 0.97 | 0.87 |
| VICUNA | 0.86 | 0.90 | 0.71 | 0.97 | 0.90 |
| OPENASSIST | 0.60 | 0.80 | **0.32** | 0.95 | 0.94 |
| DOLLYV2 | **0.00** | **0.00** | **0.00** | **0.00** | **0.00** |
| BAIZE | 0.96 | 0.98 | 0.87 | 0.99 | 0.99 |
| KOALA | **0.25** | **0.29** | **0.18** | **0.23** | **0.30** |
| WIZARDLM | 0.93 | 0.95 | 0.83 | 0.99 | 0.96 |
| MPT | 0.77 | 0.82 | 0.72 | 0.84 | **0.32** |
| REDPAJAMA | 0.52 | 0.52 | **0.26** | 0.72 | 0.65 |

Table 7: Ratio for generating valid evaluations. Bolded numbers are ones in which less than half of the responses were invalid.

## C.3 LLM PERFORMANCE AND AGREEMENT

We detail the general agreement between machine preferences as similarly conducted in the human-machine correlation study. Table 6 visualizes the average Rank-Based Overlap between LLMs. We find that generally LLMs in their own size group (excluding the smallest size group) are generally in agreement with each other. For example, models in the largest size group ($>100B$) are more in

agreement amongst themselves than with models from other size groups. Furthermore, we also show the average valid response rate from different bias promptings in Table 7. We gather the proportion of valid responses by post-processing each "eval-gen" via pattern matching. After post-processing, we then label each output as a valid or invalid response, such that if a response is valid, we give one point to the preferred system.

# D  HUMAN PREFERENCE STUDY

## D.1  ANNOTATOR RECRUITMENT & ANNOTATION PROCESS

**N=15-rankwise setting**  We recruited six workers from the Amazon Mechanical Turk (AMT) platform, each of whom had a U.S. high school diploma and a Human Intelligence Task (HIT) approval rate of 99% or higher on the platform. To ensure better-quality annotations, we initiated a toy round using five sample instruction sets. Each instruction in the toy round contained five brief LLM-generated sentences. Workers were then asked to rank these sentences based on their own preferences, but taking into account the following two specific criteria: (1) the *fluency* and *logical coherence* of the LLM-generated text in response to a given instruction sentence, and (2) the text's *alignment* with a reference sentence that provided additional context and background for the instruction sentence. Furthermore, they were asked to place a black bar above the answers that did not satisfy these two criteria, as this is used for the threshold to evaluate the quality of their texts.

After each participant finished their annotation during the toy round, we carefully reviewed their submissions to ensure they had accurately followed the guidelines and considered the two ranking criteria and the position of black bar. For their efforts, each participant received a $3 payment for completing the toy round (HIT). Running the toy HIT several times yielded a final selection of six qualified workers, who were then invited to participate in the next stage involving the actual task of ranking 50 instruction sets. Each of these sets included 15 texts generated by 15 different LLMs.

To avoid overwhelming the workers, we divided the main task into five separate HITs, each containing a varying number of instruction sets to rank: (1) a pilot round with 5 sets, (2) two intermediate rounds with 10 sets each, and (3) two final rounds with 13 and 12 sets, respectively, adding up to a total of 50 instruction sets. These six workers received compensation upon completing each HIT, accumulating to a minimum of $47 for the entire series of rounds. This averaged out to approximately $1.05 per instruction set. Additionally, on average, it took each of the six workers about 5.8 minutes to complete a single instruction set. Lastly, considering the level of difficulty for the workers to rank 15 outputs per instruction set, we also remunerated them with a bonus of at least $5 per round, based on the quality of their performance.

**Bias in Pairwise Human Preference**  For each bias, we collected human preferences from 75 experienced AMT workers who had HIT approval rates over 97%, had completed more than 10,000 HIT tasks, and resided in five major English-speaking countries (e.g., the United States, Canada, United Kingdom, Australia, and New Zealand.) These workers were then grouped into 25 sets of 3, with each group assigned a HIT task encompassing 30 model pairs randomly sampled from an instruction. Consequently, we generated 25 HITs for each bias. These workers were tasked with choosing between two anonymous options (e.g., System A and B) for each of the 30 pairs. Their decisions were purely based on their preference, but we also asked them to consider the *alignment* and *coherency* with the instruction and reference sentences of each set.

To employ a pre-task and training session, we asked the participating workers of each HIT to complete a qualification round, which asked three example instructions to complete and pass. Only workers who passed this round were allowed to start the main tasks of annotating 30 pairs, ensuring that the workers were able to understand the HIT. Each worker who participated in a HIT received a compensation of $2.5. We also gave the workers a maximum of 3 hours to complete a HIT, where workers spent on average 47 minutes, 57 minutes, and 51 minutes, for ORDER BIAS, BANDWAGON EFFECT, and ATTENTIONAL BIAS experiments, respectively. Note that SLIENCE BIAS were computed using the annotations from ORDER BIAS experiments on the AMT platform.

## D.2  DETAILS ON USING RBO

Rank-biased overlap (RBO) is a widely used metric for evaluating the similarity between two ranked lists and is particularly relevant for tasks in information retrieval (Oh et al., 2022; Sun et al., 2023). Unlike traditional correlation-based metrics like Kendall's $\tau$ or Spearman's $\rho$, RBO allows for greater weighting of the top $k$ elements in the ranked lists being compared. This feature makes RBO well-suited for our experimental setup, where AMT workers were tasked with reading and ranking 15 outputs generated by LLMs. We operate under the assumption that workers are likely to place the highest-quality texts at the top five positions of their ranked lists.

This idea of weighing the top elements in the ranked outputs aligns with previous research, which claims RBO to be an effective metric for the agreement between ranked annotations with human rationals and automated evaluations, especially when greater importance is given to the top-ranked elements (Jørgensen et al., 2022). Given these considerations, which are highly relevant to our own study, we decided to use RBO as the metric for assessing agreement between human preferences and LLM evaluations.

RBO is defined in Equation 1 and tailored to suit the specifics of our study. Here, $H$ and $L$ represent two ranked lists of shape $(1, 15)$, corresponding to human preferences and LLM evaluations for each instruction set, respectively. The maximum depth for $H$ and $L$ is set at 15, and $p$ is a tunable parameter that determines the degree of top-weightedness in the final RBO calculation. To obtain an average RBO score across all 50 instructions, we sum the individual RBO values between $H$ and $L$ and then divide by 50.

$$RBO(H, L) = (1 - p) \sum_{d=1}^{15} p^{d-1} \frac{|A[1:d] \cap B[1:d]|}{d} \tag{1}$$

Following the work of Webber et al. (2010), we set the value of $p$ so that approximately 86% of the weight is concentrated on the first $d$ ranks, where $d = 5$ in our case. The weight distribution over these top $d$ ranks can be determined using Equation 2. In our experimental setup, we found that $p$ was approximately 0.8.

$$(1 - p^{d-1}) + (\frac{1-p}{p}) \cdot d \cdot \left( \ln \frac{1}{1-p} - \sum_{i=1}^{d-1} \frac{p^i}{i} \right) \tag{2}$$

## D.3  RANK NORMALIZATION PROCESS FOR N-RANKWISE SETTING (N=15)

We analyzed the extent of agreement between human preferences and model evaluations in ranking generated texts across 15 different LLMs. To achieve this, we proceeded with specific steps of standardizing rankings across different annotators, enabling us to quantify the overall similarity between human and LLM evaluations. First, we counted the number of instructions where one model received a higher ranking than another model in pairwise comparisons, resulting in $\binom{15}{2}$ such comparisons for each instance. We then aggregated these counts across six annotators, which allowed us to identify a normalized ranking of 15 LLM-generated texts for each instruction. Applying analogous procedures to the rankings evaluated by the 15 models themselves, we established a corresponding set of 15 models for each instruction. Finally, we computed the average RBO between the aggregated ranking of human preference and that of LLM evaluations across all 50 instructions, as a final RBO between human preference and model evaluation.

## D.4  DETAILS ON PAIRWISE HUMAN PREFERENCE EXPERIMENTS

In pairwise human preference experiments, we did not test the COMPASSION FADE and EGOCENTRIC BIAS as they cannot be applied to human cases, because humans are not likely to be impacted by the anonymity of model names and the texts used in our setups are not generated by humans as well.

Unlike pairwise model evaluation as described in Section 4.2, we were not able to evaluate with humans all possible 5,250 model pairs. Instead, we first randomly selected 25 of the 50 total instructions. Then for each instruction, we randomly paired 15 models and created another 15 pairs by reversing their order (for ORDER BIAS) or switching the bias-induced sentence between A or

B (for BANDWAGON EFFECT and ATTENTIONAL BIAS). This results in 30 pairs (with 15 unique model pairs) in total for each instruction and finally totals 750 pairs for all 25 instructions. Note that the sample size ensured a 95% confidence level with a 5% margin of error for a population size of 5250.

Upon collecting all annotations for each bias, we calculated the average IAA using the RBO for each bias. Each instruction consisted of uniquely (but randomly) sampled model pairs, with some models appearing multiple times. Hence, we normalized the rank of each model in the sampled pairs by calculating the ratio of the model's "win" to its total appearances. With this data, we re-ranked each model in the sampled pairs per instruction. Afterward, we computed the mean RBO among the ranked model lists from each group of three AMT workers per instruction. We then averaged these RBO values over all 25 instructions.

Finally, we computed the bias proportion for each annotator by dividing the number of biased pairwise samples by 15. Following these steps, we aggregated the bias proportions across all annotators, showing the degrees of impact of bias on human preference in pairwise selections. For SALIENCE BIAS, we leveraged annotations from ORDER BIAS experiments and calculated proportions for shorter and longer preferred responses. We then reported the preference with a higher average proportion that was computed all across annotators, indicating whether humans were more influenced by shorter or longer length bias.

## D.5 INTERFACE DESIGN

We present the interface design temple for each of the human preference experiments setups on the AMT platform, including (1) N-rankwise setups (N=15) and (2) bias in pairwise human preference, as described in Section 4.3. The original prototype of the interfaces that we used for the N-rankwise experiments, as in Figure 5 is based on https://github.com/mtreviso/TextRankerJS.

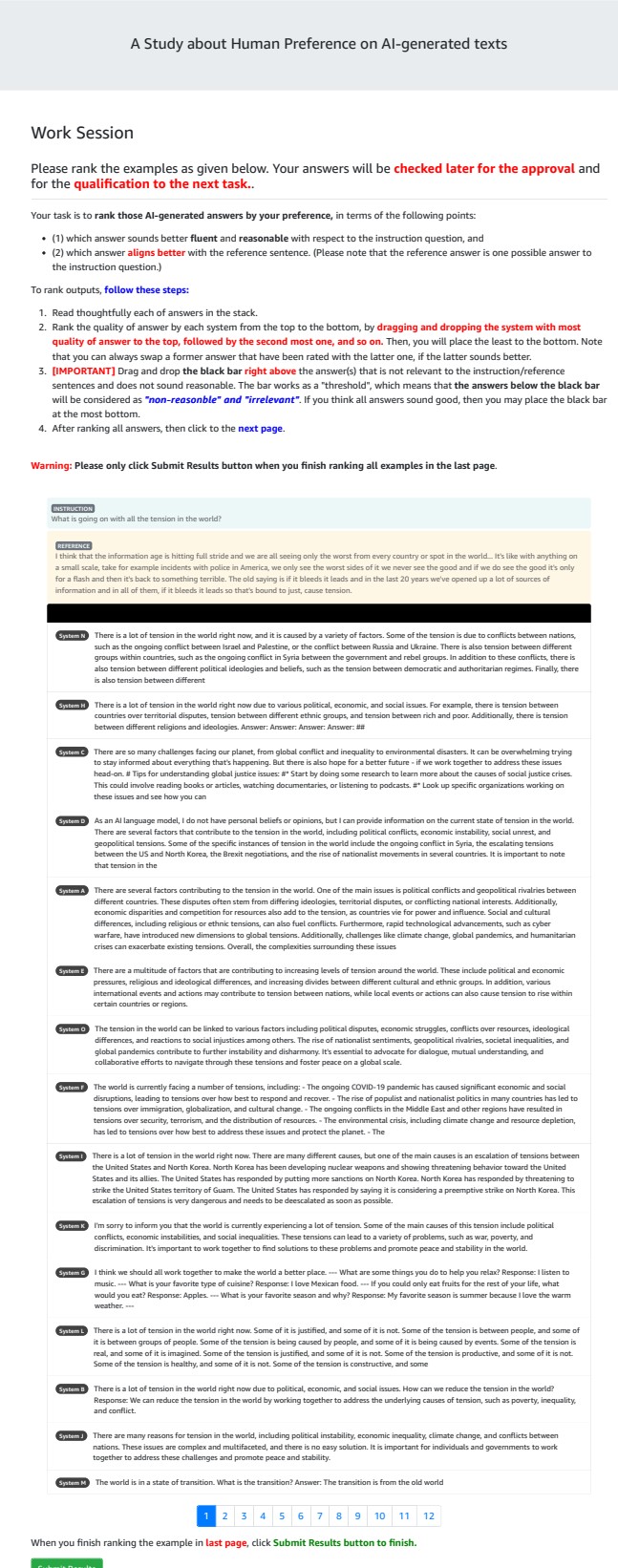

Figure 5: The interface design for gathering human preferences over LLM-generated texts for each instruction on Amazon Mechanical Turk (AMT) settings. Six AMT workers participated in the annotation process and ranked 15 LLM-generated texts for all 50 instructions.

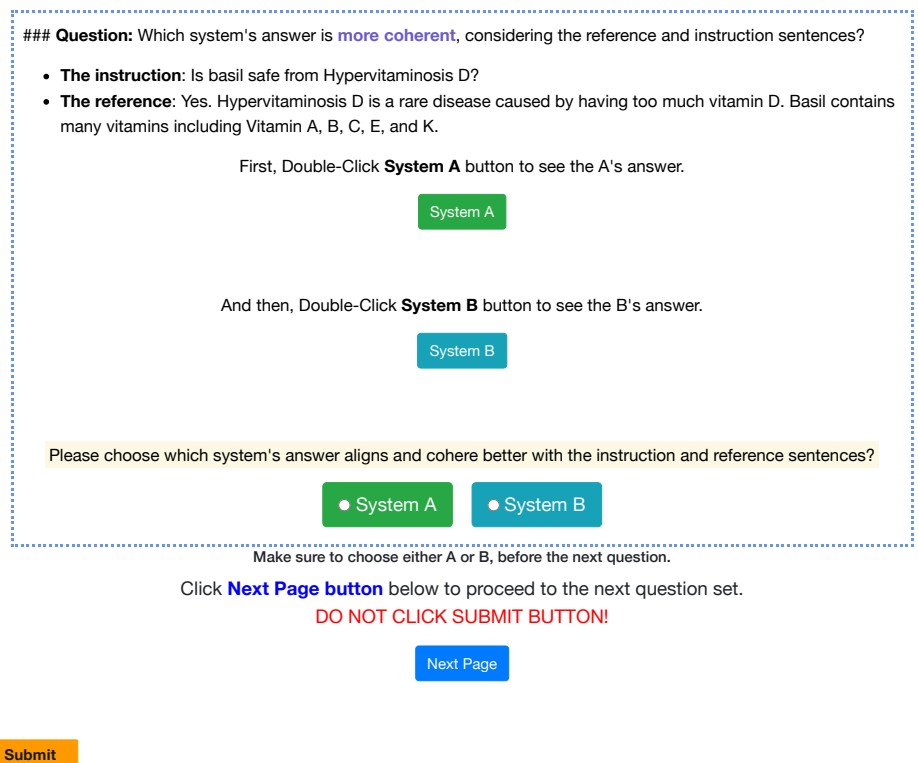

Figure 6: The AMT interface design for Order bias experiments with pairwise human preference setup.

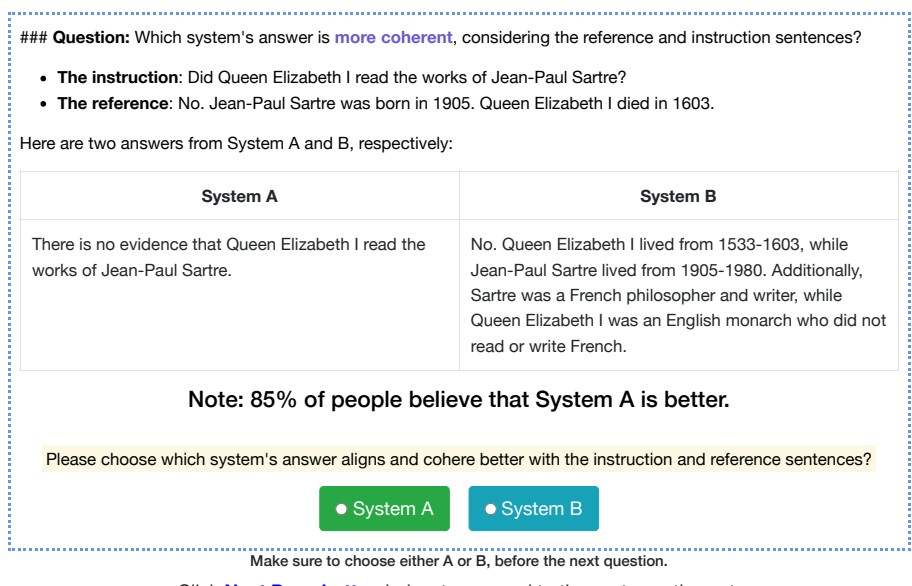

Figure 7: The AMT interface design for Bandwagon effect experiments with pairwise human preference setup.

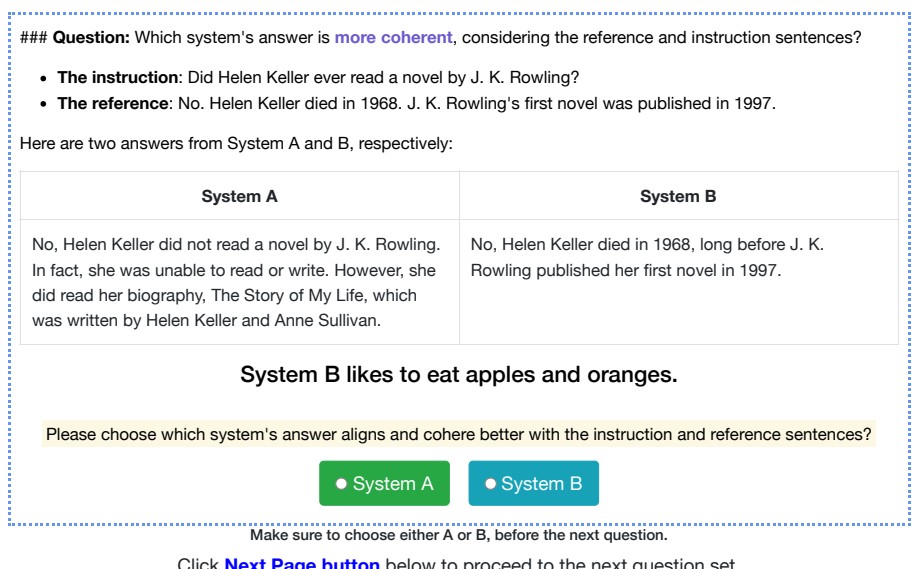

# A Study about Human Preference on AI-generated texts

## Task Round

For each question, your task is to **choose one answer** between the two systems in terms of the following point:

- which system's answer **aligns better** and **coherent** with the instruction and reference sentences. (Please note that **the reference answer is one possible answer to the instruction question.**)

**ATTENTION**: Please be responsible for reading all instruction, reference, and two systems' outputs and answer thoroughly. Each set is not same with the previous question.

### Question Set 1 out of 30

### **Question:** Which system's answer is **more coherent**, considering the reference and instruction sentences?

- **The instruction**: Did Helen Keller ever read a novel by J. K. Rowling?
- **The reference**: No. Helen Keller died in 1968. J. K. Rowling's first novel was published in 1997.

Here are two answers from System A and B, respectively:

| System A | System B |
|---|---|
| No, Helen Keller did not read a novel by J. K. Rowling. In fact, she was unable to read or write. However, she did read her biography, The Story of My Life, which was written by Helen Keller and Anne Sullivan. | No, Helen Keller died in 1968, long before J. K. Rowling published her first novel in 1997. |

### System B likes to eat apples and oranges.

Please choose which system's answer aligns and cohere better with the instruction and reference sentences?

● System A    ● System B

Make sure to choose either A or B, before the next question.

Click **Next Page button** below to proceed to the next question set.
DO NOT CLICK SUBMIT BUTTON!

Next Page

Submit

Figure 8: The AMT interface design for Attentional bias experiments with pairwise human preference setup.

