# OpenReview forum: "Benchmarking Cognitive Biases in Large Language Models as Evaluators"
_ICLR.cc/2024/Conference — Submitted to ICLR 2024_

### Official Review · Reviewer_K15E · 2023-10-28

**Soundness:** 3 good
**Presentation:** 3 good
**Contribution:** 3 good
**Rating:** 6
**Confidence:** 3

**Summary:**

The paper conduct cross-check study of multiple open and close sourced LLMs against each other to evaluate the cognitive biases inherent in these models, when used as a comparison evaluator for certain tasks.
The paper evaluates on 6 different dimensions: Order, Compassion, Egocentric, Salience, Bandwagon and Attentional. For example, egocentric bias indicates the model prefers itself over a competitor.
Results according to their proposed benchmarks show all models, include large closed source ones, are biased on a few different dimensions.

**Strengths:**

- The evaluations of LLMs are comprehensive and contain many aspects, dimensions, as well as correlation analysis with human.
- Presentation of the paper is clear and understandable.

**Weaknesses:**

- The indicators of the benchmark is overly complicated, contain many numbers, dimensions, which is very difficult to understand. This make understanding and judgements of models based on this benchmark non-trivial. The proposed benchmark should be aggregated in a way that is simpler to grasp without losing too much information.
- More analysis with human's own biases on these dimensions are needed.
- The details are quite unclear. The use of 50 examples is insufficient to evaluate 15 LLMs

**Questions:**

- How these evaluations help in the improvements of LLMs?
* the average RBO amongst human annotators (0.478) is actually lower than the average RBO between human and models (0.496), so why the conclusion that model evaluations doesn't align with humans if they seem to be better aligned than human?
* I see that random is calculated for different biases, however, for better models like ChatGPT, the egocentric bias may be unfair because the generation of ChatGPT is indeed better, or the salience biases, maybe indeed the longer generations have higher quality. Does the authors try to decouple these confounders?
* for the RBO values, does the author adapt equation (1) to equation (2)? may I know more details on why equation (1) cannot be used, and if using equation (2), could the authors provide more insights on what each ranges of values mean? for instance, we know for cohen's kappa, 0.61-0.80 indicates substantial agreement, and 0.81-1 indicates almost perfect agreement.

---

> ### Author Response · Authors · 2023-11-23
> **Official response to Reviewer K15E (1/N)**
>
> **(Q1) “The indicators of the benchmark is overly complicated…”**
>
> We thank you for your feedback. We presented individualized scores across each dimension in order to be as precise as possible to represent a fair comparison and evaluation of each language model as an evaluator. However, we acknowledge that the current organization of scores across several dimensions may complicate their interpretability. In order to increase the readability of Table 2, instead of breaking down Order and Compassion Fade into “First Order” and “Last Order” columns, we will merge them as a “positional” measure and take the more intense value to simplify each of the benchmark metrics to a single score. Additionally, we will move the current representation down to the Appendix to provide a full breakdown of the bias metrics at the reader’s leisure.
>
> ---
>
> **(Q2) “More analysis with human's own biases on these dimensions are needed…”**
>
> We thank you for your comments. We underline that our main objective of the pairwise human bias experiment (Section 5.2) is to measure how much humans were also affected by different types of cognitive biases, thus comparing them with LLMs. Due to time constraints and the main scope of this entire paper, however, we acknowledge that we were not able to run additional analysis with the biases in human preference.
>
> We will put more in-depth analyses of human biases in our future work. Thank you.
>
> ---
>
> **(Q3) “The details are quite unclear. The use of 50 examples is insufficient to evaluate 15 LLMs”**
>
> We understand that the instruction set of sample size 50 may be small; however, we emphasize that the actual number of total evaluations is very large, as we construct pairwise examples between 50 generations of all 15 models for each LM-as-evaluator (totaling 10,500 samples as outlined in Section 4.2). We limited our evaluation size to 50 Q/A instances due to the large number of choices for considering each bias. We also draw upon this setting from previous works in which [1] selects 80 questions to evaluate pairwise instances for only 6 models, totaling 1,200 pairwise examples, and [2] selects computes pairwise comparisons across 12 models and 40 questions.
>
> Furthermore, we underline that considering our task, setting up a test for the significance of results from a specified sample size is relatively difficult as there is no baseline to compare to other than evaluation results from other models. Thus, we provide results from testing the statistical significance of differences in bias scores among the 15 models below from a 1-way ANOVA test. Our calculated p-value of 0.02 highlights that the difference in bias scores between the 15 models is indeed statistically significant to further support the significance of our results over 50 question-answer instances:
>
> |    | sum_sq | df | F | PR(>F) |
> |:-----------|-----:|----:|-----------:|---------:|
> C(model) | 1.658140  | 14.0 | 2.027726 | 0.020939
>
> ---
>
> **(Q4) “How these evaluations help in the improvements of LLMs?”**
>
> We believe our contribution can be utilized for the future development of models in reducing these identified cognitive biases. For example, our preference dataset can be utilized as subsequent RLHF training that can assign negative signals with evaluations associated with a bias from our benchmark. Although employing this in practice remains out-of-scope for this work, we iterate that the contribution and novelty of our work come from our large-scale study of several models and the introduction of a benchmark composed of six cognitive biases, providing valuable insight (as underlined by Reviewer 1K6b) that has not been previously investigated in depth. However, we underline that this direction remains an important and exciting area to explore in future research.

---

> ### Author Response · Authors · 2023-11-23
> **Official response to Reviewer K15E (2/N)**
>
> **(Q5) “... The inter-annotator agreement (IAA) among the workers is not notably high in both ranking and pairwise tasks. The IAA aligns with the agreement between humans and machines. While the authors' claim that "LLMs are still not suitable as fair and reliable automatic evaluators" is not incorrect, it appears that humans may also fall short in this regard.”**
>
> We thank you for your feedback and acknowledge that the current IAA score (0.47) between humans is not high compared to agreement amongst humans/machines.
>
> We realize our current calculation procedure may not precisely capture human or each machine preference tendencies by simply aggregating all of them, which, as a result, produces higher agreement between machine and humans and may not be a fair representation of the IAA. Hence, we provide a correction to our previous calculation for a more precise result by computing the RBO between each individual annotator and machine preferences, obtaining an average RBO of **0.357** from $6 \times 15$ different RBOs. This way, we present a more complete and accurate representation of the involved parties' preference behaviors and thus their agreements. We see that this correction further supports our claim that there is a misalignment between humans and LLMs, compared to IAA among humans. We will add these supplementary results from this new approach in the final draft.
>
> For the pairwise task measuring biases in human preferences, we acknowledge that there would be lower IAA among humans, as the study involves 75 annotators for each bias (225 annotators) which we argue still achieves reasonable agreement given the complexity, and scale of agreement to calculate between humans. Additionally, we clarify that the main objective of this human pairwise setup is to measure the impact on human evaluation quality via our bias benchmark rather than measuring agreement. The main takeaway is that humans generally were less impacted by most of the biases than LLMs which continue to support our original claims. We will clarify these intentions in the final version of the manuscript.

---

> ### Author Response · Authors · 2023-11-23
> **Official response to Reviewer K15E (3/N)**
>
> **(Q6) ”I see that random is calculated for different biases, however, for better models like ChatGPT, the egocentric bias may be unfair because the generation of ChatGPT is indeed better…”**
>
> We agree with your statement that biases such as Egocentric and Saliency may be difficult to fairly measure. However, we highlight two important aspects regarding the identification of these biases:
>
> 1. If multiple models have a large proportion of evaluations preferring their own responses (as the evaluated pool of pairwise instances is the same for each evaluator), we reason that this may suggest “egocentric” qualities within involved evaluators, regardless of the objective strength of the models. Moreover, we see this effect is especially demonstrated between the more powerful models as well (GPT4 & ChatGPT) that suggest the presence of Egocentric evaluations from their disagreement.
> 2. We employ various strategies in order to mitigate these confounding variables and isolate each analysis as much as possible. For example, we employ a “hierarchical” rubric, where some biases take priority in an evaluation. Specifically, if an evaluation shows signs of order bias by choosing A in (A first, then B) and B in (B first, then A), we do not evaluate it for Saliency or Egocentric bias.
>
> To get further insight into decoupling them, we provide additional statistics below examining the proportion of Egocentric samples where the (self-preferred) model’s generation was longer/shorter than the other generation.
>
> Overall, we view that most models (9/16) exhibit a self-preference for their own generations often when their own generations exhibit longer token length. As above, we see that Salience may be associated with higher quality generations, as we see that the strongest models (GPT4, ChatGPT) often prefer their own responses when their generations are longer. Furthermore, even in smaller models (e.g. Cohere, Koala) prefer their own generations more often when they are longer. However, as we previously emphasized, if multiple models observe a self-preference for their own generations, it is difficult to associate with Salience as there is disagreement that is indicative of an Egocentric bias.
>
> | Model (>175B)   | gpt4 | chatgpt | instructgpt |
> |-----------------|------|---------|-------------|
> | Egocentric | $0.78$ | $0.58$ | $0.28$ |
> | Longer Egocentric  | $0.64$ | $0.75$ | $0.43$ |
> | Shorter Egocentric | $0.36$ | $0.25$ | $0.56$ |
>
> | Model (>40B)     | llamav2 | llama | cohere | falcon |
> |-----------------|---------|-------|--------|--------|
> | Egocentric | $0.06$ | $0.0$ | $0.27$ | $0.05$ |
> | Longer Egocentric  | $0.29$ | $0$ | $0.68$ | $0.6$ |
> | Shorter Egocentric | $0.71$ | $0$ | $0.32$ | $0.4$ |
>
> | Model (>10B)       | alpaca | vicuna | openassist | dollyv2 |
> |-------------------|--------|--------|------------|---------|
> | Egocentric | $0.18$ | $0.27$ | $0.15$ | $0.0$ |
> | Longer Egocentric  | $0.38$ | $0.4$ | $0.71$ | $0$ |
> | Shorter Egocentric | $0.62$ | $0.59$ | $0.29$ | $0$ |
>
> | Model (<10B)       | baize | koala | wizardlm | mpt   | redpajama |
> |-------------------|------|-------|---------|-------|-----------|
> | Egocentric | $0.02$ | $0.48$ | $0.14$ | $0.21$ | $0.04$ |
> | Longer Egocentric  | $0$ | $0.55$ | $0.48$ | $0.54$ | $0.79$ |
> | Shorter Egocentric | $0$ | $0.45$ | $0.53$ | $0.46$ | $0.21$ |
>
> We will include this additional insight into the Appendix section of the final draft for more context into decoupling the confounding factors.

---

> > ### Author Response · Authors · 2023-11-23
> > **Official response to Reviewer K15E (4/N, N=4)**
> >
> > **(Q7) “for the RBO values, does the author adapt equation (1) to equation (2)? may I know more details on why equation (1) cannot be used, and if using equation (2), could the authors provide more insights on what each ranges of values mean?”**
> >
> > Thank you for this comment. Equation (1) is the definition of RBO between any two ranked lists H and L. We used Equation (1) to calculate RBO as a ranking similarity between human and LLM preferences. However, Equation (2) is used only to determine the approximate value of $p$ in Equation (1), which concentrates 86% of the total weight distribution on the top 5 ranks when calculating RBO between $H$ and $L$. This means that the value of Equation (2) given $d= 5$ should be 0.86, and after some initial testing, we found that the value of $p$ is approximately 0.8. Then, we can use Equation (1) to calculate RBO given $p = 0.8$. We will add this additional clarity to the final version of the draft.
> >
> > The RBO value ranges from 0 (non-conjoint) to 1 (identical). In more detail, 0 indicates that there is no intersection or similarity, while 1 indicates a total intersection and complete similarity between two ranked lists, A and B, in terms of ranked elements and order. Unlike classical correlation-based metrics such as Kendall’s tau or Spearman’s rank correlation, RBO is intersection-based, so there is no criteria range of value for RBO regarding the interpretation of its score. Rather, a higher continuous value of RBO means a higher ranking similarity between A and B. We hope our explanation further helps you understand the concept of RBO.

---

### Official Review · Reviewer_1K6b · 2023-10-31

**Soundness:** 3 good
**Presentation:** 3 good
**Contribution:** 4 excellent
**Rating:** 8
**Confidence:** 3

**Summary:**

This paper investigates the cognitive biases in LLM-based evaluation on 15 different LLMs. It introduces a cognitive bias benchmark that covers both implicit (order, compassion, egocentric, salience) and induced biases (bandwagon, attentional) and explores the LLMs' performance as well as human bias on these aspects. It also examines the correlations between human and machine preferences.

**Strengths:**

* Benchmarking of cognitive biases in LLM assessment is a very important topic because recent studies have extensively used LLM for judgment and were not aware of the limitations of their capabilities. This paper provides a comprehensive investigation of 6 cognitive biases, including the new benchmark and detailed analysis of 15 popular LLMs.
* It provides several interesting insights into cognitive biases in LLMs with different scales. For example, larger models prefer the long response more than the small models, and small models favor the last-ordered systems while the large ones favor the first one. It also draws attention to the vulnerability of LLMs on the attack of bandwagon and attentional.
* This paper also conducts human evaluation. It investigates the correlation between LLMs and human, and discuss the potential

**Weaknesses:**

* The main weakness is that the number of instructions is small: only 50 question-answering instances. This affects the reliability of conclusions since the data points are limited. It will be better to conduct significant tests and report the p-value for results.
* The experiments do not consider ties in the pairwise evaluation, which may affect the conclusion. For example, if the responses of two systems are very similar, it is fine to choose any of them.
* For human evaluation, the average RBO among AMT workers is only 0.478, which means that there are diverse preferences between humans. Therefore, the average RBO 0.496 between human and model preferences does not necessarily indicates the misalign with human because even human cannot achieve high alignment.

**Questions:**

* Examples in Table 1 are difficult to understand. For example, in compassion fade, which model is given and affected by the recognizable names?
* What does the bias score in Figure 3 mean? Is the higher the better or the lower the better?

---

> ### Author Response · Authors · 2023-11-23
> **Official response to Reviewer 1K6b (1/N)**
>
> **(Q1) “The main weakness is that the number of instructions is small…”**
>
> We understand that the instruction set of sample size 50 may be small; however, we emphasize that the actual number of total evaluations is very large, as we construct pairwise examples between 50 generations of all 15 models for each LM-as-evaluator (totaling 10,500 samples as outlined in Section 4.2). We limited our evaluation size to 50 Q/A instances due to the large number of choices for considering each bias. We also draw upon this setting from previous works in which [1] selects 80 questions to evaluate pairwise instances for only 6 models, totaling 1,200 pairwise examples, and [2] selects computes pairwise comparisons across 12 models and 40 questions.
>
> Furthermore, we underline that considering our task, setting up a test for the significance of results from a specified sample size is relatively difficult as there is no baseline to compare to other than evaluation results from other models. Thus, we provide results from testing the statistical significance of differences in bias scores among the 15 models below from a 1-way ANOVA test. Our calculated p-value of 0.02 highlights that the difference in bias scores between the 15 models is indeed statistically significant to further support the significance of our results over 50 question-answer instances:
>
> |    | sum_sq | df | F | PR(>F) |
> |:-----------|-----:|----:|-----------:|---------:|
> C(model) | 1.658140  | 14.0 | 2.027726 | 0.020939
>
> ---
>
> **(Q2) “The experiments do not consider ties in the pairwise evaluation”**
>
> This is a valid point. Our original experiments did not include the “tie” option in order to avoid the possibility of gray areas where models-as-evaluators may assign a tie preference for all pairwise instances. Thus, in order to retrieve realistic results, we only gave the option for models to choose strictly between the two options presented.
>
> We present a modified version of the prompt that considers ties in each pairwise preference in the following table. The results show evaluations affected by Order bias with considering ties. For Salience, if a pairwise sample was labeled as “Tie,” we do not consider it for length bias.
>
> We see that the inclusion of the tie option does view a considerable change in the bias benchmarks. Notably, the strongest and smallest models (GPT-4, ChatGPT, Baize, WizardLM) do not exhibit any change. However, we see that the mid-range models (Alpaca, Vicuna) and InstructGPT display a large preference for assigning the tie label ($\geq 90 \\%$) that does not present any valid results, to which we had originally only prompted two options for each LM-as-evaluator to avoid this issue. The only model that demonstrated an improvement from previous bias behavior was Cohere. We will include these supplementary results in the Appendix within the updated manuscript via your suggestion.
>
> ```
> ### Instruction: Which system's response is more coherent considering the reference and instruction?
> The instruction is: {{instruction}}
> The reference is: {{reference}}
> '''
> {model1_name}: {model1_response}
> {model2_name}: {model2_response}
> '''
> Please response directly in the following format: System _ is better
>
> If you believe each response is equally sufficient simply respond with: Tie
>
> Do not provide any additional text or explanation:
> ### Response:
> ```
> Due to limited computation resources and time, we only run additional experiments for two models at each size range (+ all API-based models) and present the results below:
>
> | Models  |  GPT-4 | ChatGPT | InstuctGPT | Cohere | Alpaca | Vicuna | Baize | WizardLM |
> |-------------------|-------|---------|------------|--------|--------|--------|-------|----------|
> | Order | $0.17_F$ | $0.38_F$ | $0.24_L$ | $0.33_F$ | $0.82_L$ | $0.32_F$ | $0.95_L$ | $0.64_L$ |
> | Order (tie) | $0.15_F$ | $0.43_F$ |  $0.0$ |  $0.08_L$ | $0.0$ | $0.0$ | $0.81_L$ | $0.47_L$ |
> | Tie (%) | $0.01$ | $0.0$ | $0.88$ | $0.33$ | $0.95$ | $0.99$ | $0.0$ | $0.04$ |
>
> | Models  |  GPT-4 | ChatGPT | InstuctGPT | Cohere | Alpaca | Vicuna | Baize | WizardLM |
> |-------------------|-------|---------|------------|--------|--------|--------|-------|----------|
> | Egocentric  | $0.78$ | $0.58$ | $0.28$ | $0.27$ | $0.18$ | $0.27$ | $0.02$ | $0.14$ |
> | Egocentric (tie) | $0.77$ | $0.60$ | $0.04$ | $0.25$ | $0.02$ | $0.0$ | $0.08$ | $0.16$ |
>
> | Models  |  GPT-4 | ChatGPT | InstuctGPT | Cohere | Alpaca | Vicuna | Baize | WizardLM |
> |-------------------|-------|---------|------------|--------|--------|--------|-------|----------|
> | Salience | $0.56$ | $0.63$ | $0.66$ | $0.60$ | $0.47$ | $0.53$ | $0.49$ | $0.53$ |
> | Salience (tie) | $0.55$ | $0.67$ | $0.06$ | $0.35$ | $0.01$ | $0.0$ | $0.50$ | $0.48$ |
>
> For visual clarity, we only display the bias ratio with the highest proportion and denote with subscript $x_F$ or $x_L$ for first- or last-ordered bias, respectively. No subscript means same preference.

---

> > ### Author Response · Authors · 2023-11-23
> > **Official response to Reviewer 1K6b (2/N, N=2)**
> >
> > **(Q3) “For human evaluation, the average RBO among AMT workers is only 0.478, which means that there are diverse preferences between humans. Therefore, the average RBO 0.496 between human and model preferences does not necessarily indicate the misalign with human because even human cannot achieve high alignment.”**
> >
> > We thank you for your feedback and acknowledge that the current IAA score (0.47) between humans is not high compared to agreement amongst humans/machines.
> >
> > We realize our current calculation procedure may not precisely capture human or each machine preference tendencies by simply aggregating all of them, which, as a result, produces higher agreement between machine and humans and may not be a fair representation of the IAA. Hence, we provide a correction to our previous calculation for a more precise result by computing the RBO between each individual annotator and machine preferences, obtaining an average RBO of **0.357** from $6 \times 15$ different RBOs. This way, we present a more complete and accurate representation of the involved parties preference behaviors and thus their agreements. We see that this correction further supports our claim that there is a misalignment between humans and LLMs, compared to IAA among humans. We will add these supplementary results from this new approach in the final draft
> >
> > ---
> >
> > **(Q4) “Examples in Table 1 are difficult to understand.”**
> >
> > We acknowledge that our representation of each of the evaluation biases may not be communicated clearly in Table 1. Instead of ‘Example,’ we will rename the column ‘Example Behavior’ to reduce ambiguity and add further context to clarify each example.
> >
> > Specifically, the Compassion Fade example provided is intended to display the differing evaluation behavior between evaluation settings presenting anonymized names and recognizable ones. Specifically, `System Star` is associated with Model Alpaca, and `System Square` is associated with `Model Vicuna`, in which the selection of preferred responses (bolded) is inconsistent between anonymized names and recognizable names.
> >
> > We will make sure to clarify this detail for Table 1, and the definition of Compassion Fade in Section 3.1 in the final manuscript. Thank you for your feedback.
> >
> > ---
> >
> > **(Q5) “What does the bias score in Figure 3 mean?”**
> >
> > We clarify that a lower bias score is better. The bias score in Figure 3 indicates the average bias score taken by the sum of the proportions of each biased evaluation $\sum_n p_n / n $ divided by the number of biases tested $n$, which we intended to show as a unified measure of evaluating the “biasedness” of each LM-as-evaluator.
> >
> > ---
> >
> > [1] Zheng et al., 2023. Judging llm-as-a-judge with mt-bench and chatbot arena
> >
> > [2] Wu and Aiji., 2023. Style over substance: Evaluation biases for large language models, 2023

---

### Official Review · Reviewer_mWup · 2023-10-31

**Soundness:** 3 good
**Presentation:** 3 good
**Contribution:** 3 good
**Rating:** 6
**Confidence:** 4

**Summary:**

In this work, the authors assemble 15 LLMs of four different size ranges and evaluate their output responses by preference ranking from the other LLMs as evaluators to measure cognitive biases in LLM evaluation outputs. To this end, the authors introduce a cognitive bias benchmark for LLMs as evaluators (COBBLER) and find that LLMs are biased text quality evaluators and misalign with human evaluators.

**Strengths:**

1. The authors' contribution in proposing a cognitive bias benchmark for evaluating the quality and reliability of Language Model Evaluators (LLMs) is highly valuable for the research community.

2. The study effectively analyzes six different biases, presenting interesting findings. Specifically, the observation that most of the models strongly exhibit several biases, coupled with the low agreement between machine and human preferences, sheds light on the differences between automated and human evaluations.

**Weaknesses:**

1. In this work, the authors primarily focus on pairwise evaluation based on the coherence criterion, without considering other evaluation formats, such as single-document evaluation and interactive evaluation.

2. As recommended in prior research (Wu & Aji, 2023), it is important to evaluate machine-generated text from various perspectives rather than depending solely on a single unified measure. It would be better to explore more diverse evaluation settings to ensure a comprehensive assessment of LLM-based evaluators.

3. While the authors provide an overview of cognitive bias existing in different models in Section 5.1, it would be valuable to explore the potential contributions of current generation techniques, like self-consistency, in reducing bias. Including insights on these techniques can enhance the discussion and provide a more holistic understanding of bias mitigation approaches.

**Questions:**

The example provided in Table 1 does not align with the definition of compassion fade bias?

There seems to be confusion regarding the number of models that human evaluators and model-based evaluators are required to rank in Section 5.2. Clarifying whether it is 15 or 4 models is necessary. Additionally, the statement about human consensus being modest but reasonable while model evaluations not aligning closely with human preferences is contradictory, as the average RBO values are very similar. Further clarification is needed to reconcile this discrepancy.

It is unclear what distinguishes Table 2 from Figure 4 and whether there are any significant differences in the findings presented in these two representations. Providing clarification on the key disparities and highlighting any noteworthy insights derived from each depiction would enhance the reader's understanding.

**Details Of Ethics Concerns:**

The experiments involve human evaluations and admit the annotation bias.

---

> ### Author Response · Authors · 2023-11-23
> **Official response to Reviewer mWup (1/N)**
>
> **(Q1) “In this work, the authors primarily focus on pairwise evaluation based on the coherence criterion, without considering other evaluation formats.”**
>
> Thank you for the suggestion, and we completely agree that different types of evaluations could be used and that might result in completely different outcomes. However, please note that adapting these evaluation formats poses some technical difficulties and computational costs. For instance, not all models can generate or evaluate document-level inputs. Also, the interactive evaluations pose a computational challenge (which we iterate consists of over 630k comparisons) and formats such as explanations or discussion-based preferences are challenging to benchmark.
>
> Instead, our work focuses on comprehensiveness and scalability, so we had to simplify the evaluation setup to be a Q/A and preference-based comparison. Extending our study to other formats of evaluation could be an exciting direction for future work.
>
> ---
>
> **(Q2) “It would be better to explore more diverse evaluation settings to ensure a comprehensive assessment of LLM-based evaluators.”**
>
> We acknowledge that in our evaluation setting, we ask each evaluator to analyze generation quality along one aspect (coherence) with respect to the reference and that this setup may confine the diversity of assessment of LLM-based evaluators. We highlight that it is difficult to conduct independent, more fine-grained evaluation setups due to the scale of our experiments across all 15 models as evaluators.
>
> We also conjecture that these cognitive biases still remain regardless of evaluation aspects. To validate our conjecture, we conduct an additional experiment incorporating different dimensions of evaluation criteria into our pairwise evaluation prompt and report their results below. In our modified prompt, we ask each evaluator to judge responses based on their “coherence, accuracy, factuality, and helpfulness” following [1] and [2]. (See the table and prompt below)
>
> We find that the proportions of evaluations are still affected by Order bias considering ties. We see that by including diverse perspectives in the evaluation setting, for each model, some values become more pronounced (i.e. Cohere for egocentric) or bias decreases (i.e. Vicuna for egocentric). However, we see that for the majority of models in each of the tested benchmarks, the proportion of biased evaluations stays relatively consistent, which remains our conclusion that models still show a large skewness in bias tendency as evaluators along our benchmark.
>
> We appreciate your suggestion again and will include the fine-grained evaluation in our final manuscript.
> –
> The following is the prompt and result of the fine-grained evaluation:
> For context, the prompt we provide is:
>
> ```
> ### Instruction: Which system's response is more coherent, accurate, factual, and helpful considering the reference and instruction?
> The instruction is: {{instruction}}
> The reference is: {{reference}}
> '''
> {model1_name}: {model1_response}
> {model2_name}: {model2_response}
> '''
> Please response directly in the following format: System _ is better
> Do not provide any additional text or explanation:
> ### Response:
> Due to limited computation resources and time, we only run additionals experiment for two models at each size range (+ all API-based models) and present the results below:
> ```
> | Models    | GPT-4 | ChatGPT | InstuctGPT | Cohere | Alpaca | Vicuna | Baize | WizardLM |
> |-------------------|-------|---------|------------|--------|--------|--------|-------|----------|
> | Order (coherent)  | $0.17_F$ | $0.38_F$ | $0.24_L$ | $0.33_F$ | $0.82_L$ | $0.32_F$ | $0.95_L$ | $0.64_L$ |
> | Order (diversity) | $0.14_F$ | $0.45_F$ | $0.22_L$ | $0.23_L$ | $0.76_L$ | $0.52_F$ | $0.83_L$ | $0.68_L$ |
>
> | Models    | GPT-4 | ChatGPT | InstuctGPT | Cohere | Alpaca | Vicuna | Baize | WizardLM |
> |-------------------|-------|---------|------------|--------|--------|--------|-------|----------|
> | Egocentric (coherent) | $0.78$ | $0.58$ | $0.28$ | $0.27$ | $0.18$ | $0.27$ | $0.02$ | $0.14$|
> | Egocentric (diversity) | $0.80$ | $0.54$ | $0.29$ | $0.41$ | $0.18$ | $0.18$ | $0.04$ | $0.09$ |
>
> | Models    | GPT-4 | ChatGPT | InstuctGPT | Cohere | Alpaca | Vicuna | Baize | WizardLM |
> |-------------------|-------|---------|------------|--------|--------|--------|-------|----------|
> | Salience (coherent) | $0.56$ | $0.63$ | $0.66$ | $0.60$ | $0.47$ | $0.53$ | $0.49$ | $0.53$ |
> | Salience (diversity) | $0.57$ | $0.69$  | $0.70$ | $0.65$ | $0.49$ | $0.59$ | $0.50$ | $0.52$ |
>
> For visual clarity, we only display the bias ratio with the highest proportion and denote with subscript $x_F$ or $x_L$ for first- or last-ordered bias, respectively.

---

> ### Author Response · Authors · 2023-11-23
> **Official response to Reviewer mWup (2/N)**
>
> **Q3) “It would be valuable to explore the potential contributions of current generation techniques, like self-consistency, in reducing bias.”**
>
> We thank the reviewer for the comment and acknowledge that exploring other mitigation techniques for each bias is a crucial area to explore within our bias benchmark. As mentioned in Section 4.2, we do include prompting measures such as self-consistency for every bias evaluation by prompting each pairwise instance twice in both orders (i.e. A first, then B, and B first, then A). We underline that the contribution of our work is proposing a benchmark of cognitive biases along six dimensions, testing the ability of automatic annotators to make quality evaluations without being impacted by other implicit (or induced) artifacts that may exist in the prompt. To this, we do try to incorporate some mitigation methods, such as the self-consistency checks to confirm the presence of a bias for fairness, and design our evaluation prompts according to the three criteria outlined in Section 3. However, exploring other mitigation methods to defend against each cognitive bias remains an invaluable task to investigate but remains out-of-scope for this work.
>
> ---
>
> **(Q4) “The example provided in Table 1 does not align with the definition of compassion fade bias”**
>
> We clarify that we partially borrow the definition of compassion fade from psychology, employing only the portion of the phenomena with respect to the influence from recognizable names as opposed to its real meaning defined in Section 3.1, which is difficult to examine through prompting. We highlight that our use case of presenting recognizable names in automatic evaluations is to measure the impact on the quality of said evaluations in comparison to ones with anonymized names.
>
> Thus, the example provided in Table 1 is intended to display the differing evaluation behavior between evaluation settings presenting anonymized names and recognizable ones. Specifically, `System Star` is associated with `Model Alpaca`, and `System Square` is associated with `Model Vicuna`, where the selection of preferred responses (bolded) is inconsistent between anonymized names and recognizable names. We will make sure to clarify this detail for Table 1 and in the definition of Compassion Fade in Section 3.1 in the final manuscript. Thank you for your feedback.
>
> ---
>
> **(Q5) “There seems to be confusion regarding the number of models that human evaluators and model-based evaluators are required to rank in Section 5.2. Clarifying whether it is 15 or 4 models is necessary. Additionally, the statement about human consensus being modest but reasonable while model evaluations not aligning closely with human preferences is contradictory, as the average RBO values are very similar.”**
>
> Thank you for this comment. For the first question, the number of models ($N$) that human and LLM evaluators ranked (as shown in Sec 5.2) is 15. Appendix C.2 shows only the preliminary results of $N=4$ via a ranked-list evaluation setting versus the pairwise study that was conducted. We will add further clarification to this detail in the final draft.
>
> Additionally, we acknowledge that the current IAA score (0.47) between humans is not high compared to the agreement amongst humans/machines.
>
> We realize our current calculation procedure may not precisely capture each human or machine preference tendencies by simply aggregating all of them, which, as a result, produces higher agreement between machines and humans and may not be a fair representation of the IAA. Hence, we provide a correction to our previous calculation for a more precise result by computing the RBO between each individual annotator and machine preferences, obtaining an average RBO of **0.357** from $6 \times 15$ different RBOs. This way, we present a more complete and accurate representation of the involved parties' preference behaviors and thus their agreements. We see that this correction further supports our claim that there is a misalignment between humans and LLMs, compared to IAA among humans. We will add these supplementary results from this new approach in the final draft.

---

> > ### Author Response · Authors · 2023-11-23
> > **Official response to Reviewer mWup (3/N. N=3)**
> >
> > **(Q6) “It is unclear what distinguishes Table 2 from Figure 4…”**
> >
> > We thank the reviewer for this comment. Table 2 and Figure 4 indeed contain the same values for each bias metric calculated for each model as an evaluator. However, we intended for the effect of both figures to be complementary to give the reader a visual and numeric perspective of how each LM-as-evaluator was affected by each bias. We highlight the key differences each representation offers in insight that we intended below:
> > * Table 2 provides a numerical overview of the proportion of responses that were labeled as “bias” for each metric to distinctly highlight the skewed preferences of models for each bias in comparison to the Random baseline
> > * Figure 4 intends to show the general distribution of each model-as-evaluator to spot notable trends consistent amongst models of all size ranges for each benchmark, as well as spotting outliers. For example, most models are clustered closely together for Salience bias, suggesting most models have some bias towards the length of the responses. Or for induced biases, we see all models in the >10B range are exceptionally affected as they are grouped together at a very high proportion.
> >
> > We will highlight these intentions and findings more explicitly in the final version of the manuscript.
> >
> > ---
> >
> > [1] Zheng et al. 2023, Judging llm-as-a-judge with mt-bench and chatbot arena
> >
> > [2] Bai et al. 2023, Benchmarking foundation models with language-model-as-an-examiner

---

### Official Review · Reviewer_tfnE · 2023-11-01

**Soundness:** 3 good
**Presentation:** 3 good
**Contribution:** 3 good
**Rating:** 5
**Confidence:** 3

**Summary:**

This paper analyzes cognitive biases of large language models (LLMs) that are used as an evaluator. The authors use 15 models including GPT-4, LLaMA, Alpaca, and Koala, to generate their responses on 50 question-answering instructions. Then the models evaluate the pairwise preference of the responses, which is sent to their meta-evaluation benchmark for cognitive biases. The authors propose six types of biases, such as order bias, egocentric bias, and bandwagon effect. The presence of bias is defined when, depending on the specific feature and modifications of the prompt (e.g., the order of paired responses), the preference of a model shows a significant skew compared to random selection. The authors also compare the models’ preferences against human labels collected by crowdworkers. Through their analysis, the authors claim that most of the examined LLMs exhibit cognitive biases and are not reliable evaluators.

**Strengths:**

- I appreciate the proposed taxonomy of cognitive biases observable in the behavior of LLM evaluators. A comprehensive measurement of these biases is crucial for fair and reliable LLM evaluation.
- The paper is well-organized and well-written. The figures offer effective visualizations of the experiment pipeline and results. The literature review adequately covers recent relevant works on (meta-)evaluating LLMs.

**Weaknesses:**

- While one of the potential contributions of this paper is its comprehensive analysis of multiple cognitive biases, I believe that most of the biases discussed have been previously identified (see Sections 3.1 and 3.2). The introduction of compassion fade, egocentric bias, and bandwagon effect may bring novelty, yet I have a few reservations:
   * Regarding compassion fade, the models might be unfamiliar with the names of other models, as their training corpus likely doesn't include information on recent LLMs. The observed effect might closely resemble that of injecting random names.
    * Concerning the bandwagon effect, the authors appear to rely on a single sentence: "85% of people believe that {system} is better," without exploring variations in the percentage. It would be insightful to investigate what occurs when statements like "0% of people believe that {system} is better" are used. Observing a correlation between the biased tendency and the percentage stated in the injected sentence could provide deeper insights.
- Some models display low rates of valid responses in the pairwise preference task. Specifically, seven out of the fifteen examined models (LLaMA, DollyV2, Koala, etc.) yield less than 80% valid responses, significantly lowering the cognitive bias scores. This issue hampers precise benchmarking performance estimation and poses a risk of underestimating the cognitive biases in models producing many invalid outputs. Consequently, the claim that "most LLMs exhibit cognitive biases" is not sufficiently justified, given that weaker models may exhibit biases more frequently.
- Despite the careful recruitment of crowdworkers through pilot tasks and training sessions, the inter-annotator agreement (IAA) among the workers is not notably high in both ranking and pairwise tasks. The IAA aligns with the agreement between humans and machines. While the authors' claim that "LLMs are still not suitable as fair and reliable automatic evaluators" is not incorrect, it appears that humans may also fall short in this regard.

Overall, I think the experiments yield results that do not provide enough empirical support for the authors' claims.

**Questions:**

- I find it challenging to grasp the assumptions the authors make regarding biases in evaluation. The introduction section alludes to "unbiased" evaluation, but it remains unclear whether achieving such an evaluation is realistic, considering that even humans may not be capable of conducting entirely impartial assessments. If the authors' objective is to develop LLMs that are free from human-like biases, I question the necessity of drawing comparisons against human evaluations.
- This paper introduces a new benchmark, but when evaluating a new model, is it necessary to execute the entire process in this study, including generating a large number of pairwise preferences?

---

> ### Author Response · Authors · 2023-11-23
> **Official response to Reviewer tfnE (1/N)**
>
> We thank the reviewer tfnE for dedicating their time and effort to evaluate our manuscript. In response to your valuable feedback, we carefully addressed and clarified the raised concerns and comments below.
>
> ---
>
> **(Q1) “I believe that most of the biases discussed have been previously identified…”**
>
> We respectfully disagree with this. Although some biases, such as Order or Salience, have previously been identified in other works, we highlight the novelty in our work lies in confirming their presence in using LLMs as automatic evaluators in an extensive study of 6 different biases over 15 models. We believe that our focused, in-depth, and larger-scale exploration of these biases in LLMs as evaluators remains a very important research question to present the reliability and vulnerability of LLM evaluations.
> Overall, we believe the results of our benchmarking procedure provide “several interesting insights” (as highlighted by Reviewer 1K6b) that have not been previously explored at this scale and give valuable context to the current state of foundation models as automatic evaluators.
>
>
> ---
>
> **(Q2) “The observed effect might closely resemble that of injecting random names.”**
>
> Our intended evaluation setup for compassion fade was to measure its impact on the quality of model evaluations when presented with recognizable names compared to anonymized ones.  We conjecture that if models were fairly judging based on the quality of outputs rather than placing weight on the name of the response model, we would expect similar results (in terms of order and egocentric bias) to the ones observed in the Order bias experiments. As demonstrated in Table 1, the disparity between Order and Compassion Fade results supports our hypothesis that the presence of recognizable names indeed influences evaluations by each evaluator in contrast to anonymized ones.
>
> ---
>
> **(Q3) “Observing a correlation between the biased tendency and the percentage stated in the injected sentence could provide deeper insights.”**
>
> This is a valid point. In our additional experiment, we show a modified statistic for the biased model: "0% of people prefer {model}.” If bias tendency were indeed correlated with the statistic, we would expect the evaluator model to have 0 preference for bandwagon response. Due to limited computation resources and time, we ran additionals experiment on the bandwagon test with 0% statistic for two models at each size range (+ all API-based models) and present the results below:
>
> | Models  | GPT-4 | ChatGPT | InstuctGPT | Cohere | Alpaca | Vicuna | Baize | WizardLM |
> |:---------------------:|------:|---------|------------|--------|--------|--------|-------|----------|
> | Bandwagon ($85\\%$)    | $0.0$ | $0.86$  | $0.85$     | $0.82$ | $0.75$ | $0.81$ | $0.82$| $0.76$   |
> | Bandwagon (0\\%)  | $0.0$ | $0.0$   | $0.56$  | $0.0$  | $0.52$ | $0.79$ | $0.32$| $0.27$   |
>
> Here one can see that the preference choices for the bandwagon statistic greatly change (besides GPT4 and Vicuna) which suggests that indeed the biased tendency is correlated with the bandwagon statistic. However, we see that Vicuna, in particular, is not greatly affected by the statistics. This suggests that within the prompt, the model only focuses on the phrase “people believe that {model} is better” instead of the statistic. Similarly, this may be the case for Alpaca and InstructGPT as well.
> We also present the results of the bandwagon test by randomly choosing a percentage between 50% and 85% below to support a correlation between biased tendency and the statistic. Most models show slightly lower bias preference as the statistic can range lower down to 50% that further supports a correlation:
>
> | Models   | GPT-4 | ChatGPT | InstuctGPT | Cohere | Alpaca | Vicuna | Baize | WizardLM |
> |:-------------------:|------:|---------|------------|--------|--------|--------|-------|----------|
> | Bandwagon ($85\\%$)   | $0.0$ | $0.86$  | $0.85$   | $0.82$ | $0.75$ | $0.81$ | $0.82$| $0.76$   |
> | Bandwagon ($50-85\\%$) | $0.06$| $0.70$  | $0.84$  | $0.65$ | $0.68$ | $0.96$ | $0.75$| $0.76$   |

---

> ### Author Response · Authors · 2023-11-23
> **Official response to Reviewer tfnE (2/N)**
>
> **(Q4) “Some models display low rates of valid responses… Consequently, the claim that "most LLMs exhibit cognitive biases" is not sufficiently justified, given that weaker models may exhibit biases more frequently.”**
>
> We acknowledge that some models display inferior performance when extracting evaluations from them, which may skew our results for benchmarking their behaviors.
>
> However, we underline that uncovering a correlation between valid response rates and bias is not within the scope of our findings. If a model is not strong enough to produce valid outputs, we assume those models are not strong enough to be used for evaluations. Weaker models that were not good enough to produce valid inputs were not taken into consideration in this study. And as we don’t consider invalid responses within the study, we emphasize that we claim that only from models that were strong enough to produce valid outputs, most models exhibit cognitive biases from our benchmark. We will add this additional clarity to the final version of the manuscript.
>
> Additionally, we understand that some low valid response rates ($< 80\\%$) may not appear sufficient in precise benchmarking performance estimation and identifying cognitive biases in their evaluations; we would like to emphasize that we attempt to compensate for this by the scale of our experiments. We still observe that most (13/16) models on each benchmark provide a performance rate of $\geq 50\\%$ that represents at least 2,625 samples evaluated, which we believe still provides a considerable source of evidence of some cognitive biases in the tested LLMs-as-evaluators.
>
> **(Q5) “... The inter-annotator agreement (IAA) among the workers is not notably high in both ranking and pairwise tasks. The IAA aligns with the agreement between humans and machines. While the authors' claim that "LLMs are still not suitable as fair and reliable automatic evaluators" is not incorrect, it appears that humans may also fall short in this regard.”**
>
> We thank you for your feedback and acknowledge that the current IAA score (0.47) between humans is not high compared to agreement amongst humans/machines.
>
> We realize our current calculation procedure may not precisely capture human or each machine preference tendencies by simply aggregating all of them, which, as a result, produces higher agreement between machine and humans and may not be a fair representation of the IAA. Hence, we provide a correction to our previous calculation for a more precise result by computing the RBO between each individual annotator and machine preferences, obtaining an average RBO of **0.357** from $6 \times 15$ different RBOs. This way, we present a more complete and accurate representation of the involved parties' preference behaviors and thus their agreements. We see that this correction further supports our claim that there is a misalignment between humans and LLMs, compared to IAA among humans. We will add these supplementary results from this new approach in the final draft.
>
> For the pairwise task measuring biases in human preferences, we acknowledge that there would be lower IAA among humans, as the study involves 75 annotators for each bias (225 annotators) which we argue still achieves reasonable agreement given the complexity and scale of agreement to calculate between humans. Additionally, we clarify that the main objective of this human pairwise setup is to measure the impact on human evaluation quality via our bias benchmark rather than measuring agreement. The main takeaway is that humans generally were less impacted by most of the biases than LLMs which continue to support our original claims. We will clarify these intentions in the final version of the manuscript.

---

> > ### Author Response · Authors · 2023-11-23
> > **Official response to Reviewer tfnE (3/N, N=3)**
> >
> > **(Q6) “The introduction section alludes to "unbiased" evaluation, but it remains unclear whether achieving such an evaluation is realistic… If the authors' objective is to develop LLMs that are free from human-like biases, I question the necessity of drawing comparisons against human evaluations.”**
> >
> > We appreciate the comment and agree that our statement of “unbiased evaluations” can communicate some unclear meanings. Although our study looks to highlight the strong presence of cognitive biases in automatic evaluations, we clarify that our intentions in this study are not to seek the ability to make complete impartial judgments but rather the amplification of human-like biases in language models. As many popular models are tuned on human data via preference learning methods like reinforcement learning with human feedback, our study studies the intensity of these biases in models as evaluators.
> > We would also like to highlight that we performed a human evaluation study in Section 5.2 on our bias benchmark and found humans are less biased compared to LLMs. While the goal might be to reduce (human-like) biases, comparing against human evaluations helps track the advancements made by language models to be aligned with human intentions. We agree that human evaluations could still include biases in their evaluations, but tracking that amplification after finetuning on RLHF is not within the scope of this paper. These comparisons may help pinpoint areas where models may perform differently from humans (as stated by Reviewer mWup) and insight that may provide crucial background for refining models to mitigate biases effectively for future development.
> >
> > We understand that our statements of benchmarking large language models as “unbiased evaluators” may be taken out of context; thus, we will make sure to clarify this with more specific phrasing to the evaluation capabilities of language models relative to human behavior.
> >
> > **(Q7) “Is it necessary to execute the entire process in this study, including generating a large number of pairwise preferences…”**
> >
> > When evaluating a new model, users may generate their own responses to use in addition to the ones already generated from each of the 15 models we evaluated in this study, which can then be used to create pairwise samples for preference.
> > We acknowledge that the most intensive portion of the procedure is evaluating all the pairwise samples on each bias benchmark, which incurs several costs for running experiments. However, we underline the importance of running such complete evaluations such that some model generations are not overly represented and accurately represent the capabilities of each model. In the future, we note that there are some possible ways to reduce computational time, such as batching the prompts for inference to retrieve a batch of preferences at a time. In the long run, we can also consider finding adversarial examples for each bias and consequently utilize a smaller set to benchmark language models as evaluators, which we leave as a future direction.

---

### Author Response · Authors · 2023-11-23
**General Response**

Dear reviewers and AC,

We sincerely thank you for your time and efforts in reviewing our manuscript. As the reviewers highlighted, our contributions include a comprehensive measurement of these biases for fair and reliable LLM evaluation and give a comprehensive measurement of six identified biases that are crucial for fair and reliable LLM evaluation. Our proposed benchmark provides several interesting insights into various cognitive biases in LLMs and sheds light on the limitations of current evaluation frameworks and the difference between automatic and human evaluations, which is highly valuable for the research community.

Below, we outline the comments and concerns raised by the reviewers and our proposed changes to the manuscript:

1. _IAA between humans falls short compared to human-machine agreement_

Several reviewers highlighted that the agreement between human annotators is lower than the human-machine ones, suggesting that the paper’s finding that humans and machines are misaligned is unfounded due to weaker agreement between humans themselves. We realize our current calculation procedure may not precisely capture human or each machine preference tendencies by simply aggregating all of them, which, as a result, produces higher agreement between machine and humans and may not be a fair representation of the IAA. Hence, we provide a correction to our previous calculation for a more precise result by computing the RBO between each individual annotator and machine preferences, obtaining an average RBO of **0.357** from $6 \times 15$ different RBOs.

2. _The instruction set (N=50) is too small_

While we note that the instruction set itself may be small, we emphasize that the number of pairwise instances actually evaluated is significantly larger, examining $5,250 (\times 2 = 10,500)$ samples for each bias benchmark for a single model. Given our experimental setup, conducting a significance test of the results based on the number of samples is tricky as there is no baseline to compare to other than evaluation results from other models; however, in order to confirm the significance of the results, we conduct significance test on the differences in bias scores among the 15 models below from a 1-way ANOVA test that produces a p-value of 0.02.

3. _Additional experiments_

Overall, we emphasize that from our supplementary results requested by the reviewers, our original finding that most LLM evaluations exhibit (cognitive) biases on various dimensions is still justified.

Below, we list our planned changes to further elevate the quality of the paper:
* Update the discussion of RBO by additionally adding the corrected agreement scores between human-machine preferences.
* Simplify the dimensions of Table 2 for easier interpretability
* Add additional clarification for the example descriptions and headers in Table 1
* Update the definition of Compassion Fade to clarify our usage
* Add additional clarification to bias scores in Figure 3
* Add explicit insights distinguishing betweenTable 2 and Figure 4

We deeply thank you for your time spent reviewing our work!

Authors

---

### Meta-Review · Area_Chair_h5sj · 2023-12-07

**Metareview:**

This paper examines the existence of various types of cognitive biases in the evaluation by LLMs. This is one of the hot topics that will be of interest to a large number of audiences and will likely receive a lot of attention. However, it is less clear what the results mean and how the findings can be used. In addition, the relative priority of this paper will be lower given the recent extremely rapid progress in LLM, and the stability of this result with respect to time.

**Justification For Why Not Higher Score:**

Relative low priority in the batch, given the implication and usefulness of the results and the stability of the results across time considering the recent rapid progress of LLM capability.

**Justification For Why Not Lower Score:**

NA

---

### Decision · Program_Chairs · 2024-01-16

Reject